# NORMALIZING FLOWS FOR INTERVENTIONAL DENSITY ESTIMATION

## ABSTRACT

Existing machine learning methods for causal inference usually estimate quantities expressed via the mean of potential outcomes (e.g., average treatment effect). However, such quantities do not capture the full information about the distribution of potential outcomes. In this work, we estimate the density of potential outcomes after interventions from observational data. For this, we propose a novel, fully-parametric deep learning method called *Interventional Normalizing Flows*. Specifically, we combine two normalizing flows, namely (i) a teacher flow for estimating nuisance parameters and (ii) a student flow for a parametric estimation of the density of potential outcomes. We further develop a tractable optimization objective based on a one-step bias correction for an efficient and doubly robust estimation of the student flow parameters. As a result our *Interventional Normalizing Flows* offer a properly normalized density estimator. Across various experiments, we demonstrate that our *Interventional Normalizing Flows* are expressive and highly effective, and scale well with both sample size and high-dimensional confounding. To the best of our knowledge, our *Interventional Normalizing Flows* are the first fully-parametric, deep learning method for density estimation of potential outcomes.

## 1 INTRODUCTION

Causal inference increasingly makes use of machine learning methods to estimate treatment effects from observational data (e.g., van der Laan et al., 2011; Künzel et al., 2019; Curth & van der Schaar, 2021; Kennedy, 2022). This is relevant for various fields including medicine (e.g., Bica et al., 2021), marketing (e.g., Yang et al., 2020), and policy-making (e.g., Hünermund et al., 2021). Here, causal inference from observational data promises great value, especially when experiments for determining treatment effects are costly or even unethical.

The vast majority of the machine learning methods for causal inference estimate *averaged* quantities expressed by the (conditional) mean of potential outcomes. Examples of such quantities are the average treatment effect (ATE) (e.g., Shi et al., 2019; Hatt & Feuerriegel, 2021), the individual treatment effect (ITE) (e.g., Shalit et al., 2017; Hassanpour & Greiner, 2019; Zhang et al., 2020), and treatment-response curves (e.g., Bica et al., 2020; Nie et al., 2021). Importantly, these estimates only describe averages *without* distributional properties.

However, making decisions based on averaged causal quantities can be misleading and, in some applications, even dangerous (Spiegelhalter, 2017; van der Bles et al., 2019). On the one hand, if potential outcomes have different variances or number of modes, relying on the average quantities provides incomplete information about potential outcomes, and may inadvertently lead to local – and not global – optima during decision-making. On the other hand, distributional knowledge is needed to account for uncertainty in potential outcomes, and thus informs how likely a certain outcome is. For example, in medicine, knowing the distribution of potential outcomes is highly important (Gische & Voelkle, 2021): it gives the probability that the potential outcome lies in a desired range, and thus defines the probability of treatment success or failure. Motivated by this, we aim to estimate the *density* of potential outcomes.

An example highlighting the need for estimating the density of potential outcomes is shown in Fig. 1. Here, we simulated outcomes according to a given structural causal model (SCM). The potential outcomes $Y[a]$ can be sampled by setting the treatment to specific value in the equation for $A$ (cf.

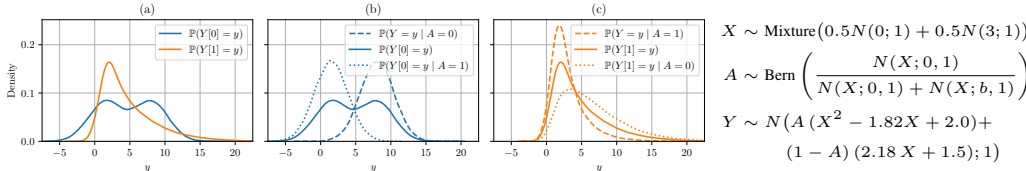

Figure 1: Motivating example showing the densities of observational, interventional, and counterfactual distributions of outcome $Y$. These are simulated via the structural causal model on the right (here: $N(x; \mu, \sigma^2)$ are densities of the normal distribution; and $b = 3$ is a covariates shift, which regulates the probability of treatment assignment). Potential outcomes have different distributions but the same mean $\mathbb{E}(Y[0]) = \mathbb{E}(Y[1]) \approx 4.77$ and the same variance $\text{var}(Y[0]) = \text{var}(Y[1]) \approx 4.06$. Here, $Y[a]$ is the potential outcome given treatment $a$. **(a)** Interventional distributions. **(b)** and **(c)** Observational and counterfactual distributions for the same outcomes. As shown here, the observational, interventional, and counterfactual distributions can be vastly different.

Appendix B). At the same time, by flipping the treatment assignment in this equation, we obtain counterfactual outcomes $Y[a] \mid A = a'$. We observe that the potential outcomes have the same mean (i.e., $\mathbb{E}(Y[0]) = \mathbb{E}(Y[1])$) and the same variance (i.e., $\text{var}(Y[0]) = \text{var}(Y[1])$). Hence, the ground-truth ATE equals zero. Nevertheless, the distributions of potential outcomes (i.e., $\mathbb{P}(Y[a])$) are clearly different. Hence, in medical practice, acting upon the ATE without knowledge of the distributions of potential outcomes could have severe, negative effects. To show this, let us consider a "do nothing" treatment ($a = 0$) and some medical treatment ($a = 1$). Further, let us consider an outcome to be successful if some risk score $Y$ is below the threshold of five. Then, the probability of treatment success (i.e., $\mathbb{P}\{Y[1] < 5.0\} \approx 0.63$) is much larger than the probability of success after the "do nothing" treatment (i.e., $\mathbb{P}\{Y[0] < 5.0\} \approx 0.51$), highlighting the importance of treatment.

In this paper, we aim to estimate the *density* of potential outcomes after intervention $a$, i.e., $\mathbb{P}(Y[a] = y)$. From this point on, we refer to this task as **interventional density estimation** (IDE). Estimating the density of interventions has several crucial advantages: it allows to identify multi-modalities in the distribution of potential outcomes; it allows to estimate quantiles of the distribution; and it allows to compute the probability with which a potential outcome lies in a certain range. Importantly, traditional density estimation methods are **not** applicable for IDE due to the fundamental problem of causal inference: that is, the counterfactual outcomes are typically never observed, and, hence, the sample from ground-truth interventional distribution is also inaccessible.

In prior literature, Kennedy et al. (2021) introduced a theory for efficient semi-parametric IDE estimation, but without a flexible algorithmic instantiation in form of a method. Existing literature also offers some specific methods for IDE, which are either semi- or non-parametric. [1] Examples are kernel density estimation (Kim et al., 2018) and kernel mean embeddings of distributions (Muandet et al., 2021). However, both methods neither scale well with the sample size nor with the dimensionality of covariates. Furthermore, both methods have an additional, crucial limitation: estimated densities could be unnormalized or even return negative values (which, by definition, is not possible). Fully-parametric methods, on the other hand, have several practical advantages: they automatically provide properly normalized density estimators, they allow one to sample from the estimated density and typically scale well with large and high-dimensional datasets. However, to the best of our knowledge, there is no fully-parametric, deep learning method for IDE.

In this paper, we develop a novel, fully-parametric deep learning method: ***Interventional Normalizing Flows*** (INFs). Our INFs build upon normalizing flows (NFs) (Tabak & Vanden-Eijnden, 2010; Rezende & Mohamed, 2015), but which we carefully adapt for causal inference. This requires several non-trivial adaptations. Specifically, we combine two NFs: a (i) teacher flow for estimating nuisance parameters, and a (ii) student flow for a parametric estimation of the density of potential outcomes. Here, we construct a novel, tractable optimization objective based on a one-step bias correction to allow for an efficient and doubly robust estimation. At the end, we develop a two-step training procedure to train both the teacher and the student flows.

Overall, our **main contributions** are following:[2]

---

[1] We distinguish the interventional distribution (i.e., $\mathbb{P}(Y[a])$) and the counterfactual distribution (i.e., $\mathbb{P}(Y[a] \mid A = a')$), which are different in general. This can be seen by comparing plots (a) vs. (b) and (c) in Fig. 1. For further information, we refer to Appendix B.

[2] Code is available at https://anonymous.4open.science/r/AnonymousInterFlow-E2F3.

1. We introduce the first fully-parametric, deep learning method for interventional density estimation, called *Interventional Normalizing Flows* (INFs). Our INFs provide a properly normalized density estimator.

2. We derive a tractable optimization problem with a one-step bias correction for efficient and doubly robust estimation. To solve, we propose a two-step training procedure with our INFs.

3. We demonstrate in various experiments that our INFs are highly expressive and effective. A major advantage owed to the parametric form of the student flow is that our INFs scale well to both large and high-dimensional datasets in comparison to other non- and semi-parametric methods.

## 2 RELATED WORK

Recently, there has been a great interest in using machine learning and, specifically, deep learning for estimating causal quantities. Examples are machine learning for estimating ATEs (e.g., Shi et al., 2019; Hatt & Feuerriegel, 2021), ITEs (e.g., Johansson et al., 2016; Alaa & van der Schaar, 2018; Wager & Athey, 2018; Curth & van der Schaar, 2021), and treatment-response curves (e.g., Bica et al., 2020; Schwab et al., 2020; Nie et al., 2021). In this regard, some papers proposed uncertainty-aware methods, e. g., by using the variance of potential outcomes (Alaa & van der Schaar, 2017; Jesson et al., 2020), or the conditional outcome distribution (Jesson et al., 2021; 2022). However, the aforementioned works are all concerned with estimating *averaged* causal quantities expressed via the mean of potential outcomes. In contrast, there are only a few papers that estimate the *density* of outcomes after intervention.

### 2.1 INTERVENTIONAL DENSITY ESTIMATION

Kennedy et al. (2021) introduced a theory for efficient semi-parametric estimation. The theory also lends to a hypothetical estimator as a solution to an integral equation, namely a bias-corrected moment condition. However, the theory comes without an algorithmic instantiation in form of a method. We later adopt the theoretical framework and convert the bias-corrected moment condition into a tractable optimization objective, which we can then solve very effectively with deep learning.

Table 1 lists existing methods for IDE. Importantly, these are either non-parametric or semi-parametric. Kim et al. (2018) developed a doubly robust kernel density estimation (KDE) via an efficient estimation of density functionals. Muandet et al. (2021) proposed kernel mean embeddings of distributions (DKME), which provides a non-parametric plug-in estimator. However, both methods (Kim et al., 2018; Muandet et al., 2021) have limitations. First, they do not provide a properly normalized density estimator. Hence, the estimated densities can be unnormalized or even negative, yet which, by definition, is not possible. Second, they do not offer direct sampling, which would allow one to sample from the estimated density without an additional algorithm. This may complicate computations of the test log-probability or empirical Wasserstein distance during evaluation. Third, another limitation of both non-parametric and semi-parametric methods is that they typically scale not well. This is unlike fully-parametric methods, which scale well to both large and high-dimensional datasets. However, so far, there is no full-parametric, deep learning method for IDE.

The methods for IDE above (Kim et al., 2018; Muandet et al., 2021; Kennedy et al., 2021) build upon general assumptions for causal identifiability. We later adopt the *same* assumptions for IDE (see Section 3), and we then develop a fully-parametric, deep learning method called INFs. Our method has three favorable properties: it yields a proper density estimator, it allows for direct sampling, and it scales well.

Table 1: Overview of methods for interventional density estimation from observational data.

| Method | Density model | Parametric | Estimator type | Proper density | Direct sampling |
|---|---|---|---|---|---|
| Kim et al. (2018) | kernel density estimation (KDE) | semi-parametric | A-IPTW | ✗ | ✗ |
| Muandet et al. (2021) | distributional kernel mean embeddings (DKME) | non-parametric | plug-in | ✗ | ✗ |
| INFs (this paper) | normalizing flows (NFs) | fully-parametric | A-IPTW | ✓ | ✓ |

A-IPTW: augmented inverse propensity of treatment weighted

### 2.2 EFFICIENT ESTIMATION

In the context of treatment effect estimation, so-called augmented inverse propensity of treatment weighted (A-IPTW) estimators were developed for efficient, semi-parametric estimation of target estimands (parameters) (Robins, 2000). Formally, A-IPTW estimation performs a first-order bias correction of plug-in models (Bickel et al., 1993; Chernozhukov et al., 2018). A-IPTW estimation

also offers the property of being double robust, i. e., fast convergence rates even if one of the nuisance parameter estimators converges slowly Kennedy (2020). Although Kennedy et al. (2021) formulated an integral equation for semi- and fully-parametric efficient IDE estimation (see Eq. 19 therein), no flexible algorithmic instantiations in form of a method have been implemented so far. Later, we reformulate this equation as a tractable optimization problem, and, thereby, turn our INFs into an efficient and doubly robust IDE estimator.

## 2.3 NORMALIZING FLOWS

Normalizing flows were introduced for expressive variational approximations in variational autoencoders (Tabak & Vanden-Eijnden, 2010; Rezende & Mohamed, 2015). One practical benefit of NFs is that they yield universal density approximators (Dinh et al., 2014; 2017; Huang et al., 2018; Durkan et al., 2019). Furthermore, NFs can be leveraged for conditional density estimation (e. g., via so-called hypernetworks (Trippe & Turner, 2018)). Normalizing flows were used for causal inference, but in a different setting from ours (see Appendix A). We provide a background on normalizing flows in Appendix B.

**Research gap:** Existing methods for IDE are either non- or semi-parametric. To the best of our knowledge, our work is the first to propose a fully-parametric, deep learning method for IDE.

## 3 SETUP: INTERVENTIONAL DENSITY ESTIMATION

**Notation.** Let $\mathbb{P}(Z)$ be a distribution of a random variable $Z$, and let $\mathbb{P}(Z = z)$ be its density or probability mass function. Let $\pi_a(x) = \mathbb{P}(A = a \mid X = x)$ denote the propensity score. Further, $\mathbb{1}(\cdot)$ is the indicator function; $\mathbb{P}_n\{f(X)\} = \frac{1}{n}\sum_{i=1}^{n} f(X_i)$ is the sample average of a random $f(X)$; and $\mathbb{P}_b^{\mathcal{B}}\{f(X)\}$ is the average evaluated on a minibatch $\mathcal{B}$ of size $b$. For readability, we sometimes highlight random variables and the corresponding averaging operator in green color. Furthermore, $\mathbb{P}(Y \mid X, A)$ is the conditional distribution of the outcome $Y$.

**Problem statement.** In this work, we aim at estimating the interventional density from observational data, namely $\hat{\mathbb{P}}(Y[a] = y)$. To compare the goodness-of-fit of different estimators, we evaluate the distributional distance between the ground-truth interventional density and the estimated density. Such distributional distances include, e.g., the average log-probability and the empirical Wasserstein distance.

We build upon the standard setting of potential outcomes framework (Rubin, 1974), where $Y[a]$ stands for the potential outcome after intervening on treatment by setting it to $a$. That is, we consider an observational sample $\mathcal{D}$ with $d_X$-dimensional covariates $X \in \mathcal{X} \subseteq \mathbb{R}^{d_X}$, a treatment $A \in \{0, 1\}$, and a $d_Y$-dimensional continuous outcome $Y \in \mathcal{Y} \subseteq \mathbb{R}^{d_Y}$, drawn i.i.d. We consider $d_Y = 1$ if not stated explicitly. We assume the treatment to be binary, but note that our INFs also work with categorical treatments. We denote $\mathcal{D} = \{X_i, A_i, Y_i\}_{i=1}^{n} \sim \mathbb{P}(X, A, Y)$, where $n$ is the sample size, and $i$ is the index of an observation. For example, in critical care, the patient covariates $X$ are different risk factors (e.g., age, gender, weight, prior diseases), the treatment is whether a ventilator is applied, and the outcome is the probability of patient survival. The covariates $X$ are also called confounders if $\mathbb{P}(Y[a]) \neq \mathbb{P}(Y \mid A = a)$.

**Identifiability.** To identify the interventional density, we make the following identifiability assumptions with respect to the data-generating mechanism of $\mathcal{D}$: (1) *Positivity:* For some $\epsilon > 0$, $\mathbb{P}\{1 - \epsilon \geq \pi_a(X) \geq \epsilon\} = 1$. (2) *Consistency:* If $A = a$ for some patient, then $Y = Y[a]$. (3) *Exchangeability:* $A \perp\!\!\!\perp Y[a] \mid X$ for all $a$. Note that these assumptions are standard in the literature (Kim et al., 2018; Kennedy et al., 2021; Muandet et al., 2021). Under assumptions (1)–(3), the density of interventional distribution $\mathbb{P}(Y[a])$ can be expressed in terms of observational distribution with back-door adjustment, i.e.,

$$\mathbb{P}(Y[a] = y) = \int_{x \in \mathcal{X}} \mathbb{P}(Y = y \mid X = x, A = a)\,\mathbb{P}(X = x)\,\mathrm{d}x = \mathop{\mathbb{E}}_{X \sim \mathbb{P}(X)}\big(\mathbb{P}(Y = y \mid X, A = a)\big), \quad (1)$$

where $\mathbb{P}(Y = y \mid X, A)$ is the conditional density of the outcome. For more details on the potential outcomes framework and identifiability, we refer to Appendix B.

**Plug-in estimator.** A straightforward approach for IDE (Robins & Rotnitzky, 2001) is the following: first, one estimates the conditional outcome distribution, $\hat{\mathbb{P}}(Y \mid X, A)$ (here, any method for

Figure 2: Overview of *Interventional Normalizing Flows*. Our INFs combine two normalizing flows, which we call "teacher flow" and "student flow". The teacher flow estimates the nuisance parameters, i.e., the propensity score $\hat{\pi}_a(X)$ and the conditional outcome distribution $\hat{\mathbb{P}}(Y \mid X, A)$. The student flow utilizes them to estimate the projection parameters $\hat{\beta}_a^{\text{A-IPTW}}$. Both teacher and student flows are fitted via a two-step training procedure.

conditional density estimation could be used). Then, one takes a sample average over covariates $X$:

$$\hat{\mathbb{P}}^{\text{PI}}(Y[a] = y) = \mathbb{P}_n\{\hat{\mathbb{P}}(Y = y \mid X, A = a)\}. \tag{2}$$

This estimator is an unbiased but inefficient estimator of interventional density, which is known as *semi-parametric plug-in estimator*. Semi-parametric IDE, unlike, e.g., semi-parametric ATE estimation, is highly problematic. For large sample sizes, the semi-parametric estimator requires averaging over the full sample for each evaluation point. Motivated by this, we aim to develop a fully-parametric estimator.

## 4 THEORETICAL BACKGROUND FOR FULLY-PARAMETRIC IDE

In this section, we introduce a theory for fully-parametric estimation of interventional density. First, we describe a parametric plug-in estimator as a solution to the moment condition (Kennedy et al., 2021). We call this estimator *covariate-adjusted estimator*. Second, we develop a one-step bias correction for efficient estimation.

We start by defining a parametric model, $\big\{g(y; \beta_a) \mid \beta_a \in \mathbb{R}^d\big\}$, where $\beta_a \in \mathbb{R}^d$ are parameters of estimator, and $g(\cdot; \beta_a)$ is a density, i.e., $\int_{y \in \mathcal{Y}} g(y; \beta_a) \, \mathrm{d}y = 1$. For IDE, we approximate the interventional distribution $\mathbb{P}(Y[a])$ with a distribution from our parametric model. That is, we aim at minimizing the distributional distance (specifically KL-divergence) between $\mathbb{P}(Y[a])$ and $g(\cdot; \beta_a)$ via

$$\hat{\beta}_a = \underset{\beta_a}{\arg\min} \, \text{KL}\left(\mathbb{P}(Y[a]) \big\| g(\cdot; \beta_a)\right) = \underset{\beta_a}{\arg\min} \, \underset{Y^a \sim \mathbb{P}(Y[a])}{\mathbb{E}}\left(-\log g(Y^a; \beta_a)\right), \tag{3}$$

where $\hat{\beta}_a$ are called *projection parameters* as they project the true interventional density onto a class $\{g(\cdot; \beta_a); \beta_a \in \mathbb{R}^d\}$.

### 4.1 COVARIATE-ADJUSTED ESTIMATOR

Let the $d$-dimensional random variable $T(Y; \beta_a) = -\nabla_{\beta_a} \log g(Y; \beta_a)$ denote the score function. Following Kennedy et al. (2021), the projection parameters can be equivalently expressed as a solution to the *moment condition* $m(\beta_a) \overset{!}{=} 0$, where

$$m(\beta_a) = \underset{Y^a \sim \mathbb{P}(Y[a])}{\mathbb{E}} T(Y^a; \beta_a) = \underset{X \sim \mathbb{P}(X)}{\mathbb{E}}\left(\mathbb{E}\big(T(Y; \beta_a) \mid X, A = a\big)\right). \tag{4}$$

Here, the moment condition is the expected score function of the potential outcome. Throughout the paper, we assume that the moment condition has a unique solution, and, therefore, the minimization task in Eq. (3) and the root-finding task in Eq. (4) are equivalent.

In practice, we have neither observations from the interventional distribution nor counterfactual outcomes. Therefore, we cannot use the ground-truth $\mathbb{P}(Y[a])$ but, instead, must use the plug-in estimator distribution from Eq. (2). Specifically, we can obtain a plug-in estimator of projection parameters, i.e., $\hat{\beta}_a^{\text{PI}}$, either by minimizing a cross-entropy loss or by solving the moment condition, both of which are equivalent:

$$\hat{\beta}_a^{\text{PI}} = \underset{\beta_a}{\arg\min} \, \underset{Y^a \sim \mathbb{P}_n\{\hat{\mathbb{P}}(Y|X, A=a)\}}{\mathbb{E}} - \log g(Y^a; \beta_a) \iff \hat{m}^{\text{PI}}(\beta_a) = \underset{Y^a \sim \mathbb{P}_n\{\hat{\mathbb{P}}(Y|X, A=a)\}}{\mathbb{E}} T(Y^a; \beta_a) \overset{!}{=} 0. \tag{5}$$

Then, we can define a *parametric covariate-adjusted* (CA) estimator as $\hat{\mathbb{P}}^{\text{CA}}(Y[a] = y) = g(y; \hat{\beta}_a^{\text{PI}})$. By choosing a sufficiently expressive class of densities for both $g$ and the conditional density estimator $\hat{\mathbb{P}}(Y \mid X, A)$ (e. g., normalizing flows), CA can be shown to consistently estimate the interventional density (see Appendix B.5 in Kennedy et al. (2021)).

## 4.2 EFFICIENT ESTIMATION VIA ONE-STEP BIAS CORRECTION

In the following, we aim to develop an efficient estimator of the projection parameter $\hat{\beta}_a$ from Eq. (3) or, equivalently, the moment condition $\hat{m}(\beta_a)$ at fixed $\beta_a$ from Eq. (4). For this, we make use of semi-parametric efficiency theory (van der Laan & Robins, 2003; Kennedy et al., 2021). We provide a background on efficiency theory in Appendix B.

Kennedy (2022) showed that the efficient influence function $\phi_a(T, \mathbb{P})$ for the functional $\mathbb{E}(\mathbb{E}(T \mid X, A = a))$ equals to

$$\phi_a(T; \mathbb{P}) = \frac{\mathbb{1}(A = a)}{\pi_a(X)}\Big(T - \mathbb{E}(T \mid X, A = a)\Big) + \mathbb{E}(T \mid X, A = a) - \underset{X \sim \mathbb{P}(X)}{\mathbb{E}}(\mathbb{E}(T \mid X, A = a)). \quad (6)$$

Here, we use red color to show the nuisance parameters of $\mathbb{P}$ that are influencing the functional. We emphasize that the nuisance parameters (i. e., the propensity score and conditional expectations/probabilities) can be either known or estimated.

The efficient influence function in Eq. (6) allows us to construct an efficient estimator of the moment condition. Following (Kennedy et al., 2021), we transform the plug-in estimator $\hat{m}^{\text{PI}}(\beta_a)$ from Eq. (5) into an efficient estimator with the help of a *one-step bias correction*. In our case, the bias-corrected moment condition has the following form:

$$\hat{m}^{\text{A-IPTW}}(\beta_a) = \hat{m}^{\text{PI}}(\beta_a) + \mathbb{P}_n\{\phi_a(T(Y; \beta_a); \hat{\mathbb{P}})\} \overset{!}{=} 0, \quad (7)$$

where $\hat{\mathbb{P}} = \{\hat{\pi}_a(x), \hat{\mathbb{P}}(Y \mid X, A)\}$ are the estimated nuisance parameters of $\mathbb{P}$. The estimated nuisance parameters are simultaneously used for plug-in estimation of the moment condition. We call the solution of the bias-corrected moment equation $\hat{\beta}_a^{\text{A-IPTW}}$ an *augmented inverse propensity of treatment weighted* (A-IPTW) estimator of the projection parameters. Unlike CA estimator, A-IPTW estimator achieves efficiency and possesses a double robustness property.

We now transform the bias-corrected moment condition into the following tractable optimization task (see Appendix C for details):

$$\hat{\beta}_a^{\text{A-IPTW}} = \underset{\beta_a}{\arg\min}\Bigg[\underbrace{\underset{Y^a \sim \mathbb{P}_n\{\hat{\mathbb{P}}(Y|X,A=a)\}}{\mathbb{E}}\Big(-\log g(Y^a; \beta_a)\Big)}_{\text{cross-entropy loss}}$$

$$+ \underbrace{\mathbb{P}_n\bigg\{\frac{\mathbb{1}(A = a)}{\hat{\pi}_a(X)}\Big(-\log g(Y; \beta_a) + \underset{Y \sim \hat{\mathbb{P}}(Y|X,A=a)}{\mathbb{E}}\big(\log g(Y; \beta_a)\big)\Big)\bigg\}}_{\text{one-step bias correction}}\Bigg]. \quad (8)$$

Previously, Kennedy et al. (2021) proposed to directly solve bias-corrected moment condition, i. e., a system of nonlinear equations, yet which is in general much harder to solve, even computationally. In contrast, we develop an optimization objective that can be directly incorporated into a loss of a deep learning density estimator.

## 5 INTERVENTIONAL NORMALIZING FLOWS

In the following, we describe our *Interventional Normalizing Flows*: a fully-parametric method for interventional density estimation via deep learning. First, we describe all the components of our architecture and, then, introduce an efficient estimation using one-step bias correction.

## 5.1 COMPONENTS

In our INFs, we combine two normalizing flows, which we refer to as (i) *teacher flow* and (ii) *student flow (see Fig. 2)*. The rationale for this is based on our derivations in Section 4, according to which a fully-parametric IDE requires two models: (i) one for the estimation of nuisance parameters, and (ii) one for the subsequent optimization of the learning objective with respect to projection parameters.

Accordingly, both NFs in our INFs have thus different objectives: (i) the teacher flow estimates the nuisance parameters (i.e., the propensity score and the conditional outcome distribution); and (ii) the student flow uses the estimated nuisance parameters to estimate the projection parameters.

**(i) Teacher flow.** The teacher flow has three components: two fully-connected (FC) subnetworks and a conditional normalizing flow parameterized by $\theta$. The first FC subnetwork (FC$_1$) takes the covariates $X$ as input and, then, outputs a representation $R \in \mathbb{R}^r$ together with a propensity score $\hat{\pi}_a(X)$. The second FC subnetwork (FC$_2$) takes the representation $R$ and the observed treatment $A_i$ as input and, then, outputs the parameters of flow, conditioned on $X$ and $A$, i.e., $\theta(X, A)$. Together, FC$_1$ and FC$_2$ form a so-called hypernetwork (Ha et al., 2017) for the conditional normalizing flow, which allows us to learn the conditional outcome distribution via back-propagation.[3]

Let $\mathcal{L}_t$ be the loss of the teacher flow. Here, we combine a conditional negative log-likelihood ($\mathcal{L}_{\mathrm{NLL}}$) and binary cross-entropy loss for the propensity score ($\mathcal{L}_\pi$), i.e., $\mathcal{L}_t(\hat{\mathbb{P}}, \hat{\pi}_a) = \mathbb{P}_n\{\mathcal{L}_{\mathrm{NLL}} + \alpha \mathcal{L}_\pi\}$ with $\mathcal{L}_{\mathrm{NLL}} = -\log \hat{\mathbb{P}}(Y = Y \mid X, A)$; $\mathcal{L}_\pi = \mathrm{BCE}(\hat{\pi}_A(X), A)$, where $\alpha > 0$ is a hyperparameter. In general, conditional normalizing flows are prone to overfitting when trained via a conditional negative log-likelihood. To address this, we later employ noise regularization (Rothfuss et al., 2019) in the conditional density estimation.

**(ii) Student flow.** The student flow uses the outputs of the teacher flow and then learns the interventional distribution. We first describe the naïve variant of the student flow without one-step bias correction (we introduce this later in Section 5.2). Different from the conditional normalizing flow in the teacher flow, the student flow is a non-conditional normalizing flow, parameterized by $\beta_a$. Specifically, we consider two separate normalizing flows, that is, one for each potential outcome (i.e., $a = 0$ and $a = 1$, respectively).[4]

To fit the student flow, we must solve the moment condition from Eq. (5) or, equivalently, minimize a cross-entropy loss ($\mathcal{L}_{\mathrm{CE}}$). Here, we use a tractable approximation via numeric integration:

$$
\mathcal{L}_{\mathrm{CE}}(\beta_a) = \mathop{\mathbb{E}}_{Y^a \sim \mathbb{P}_n\{\hat{\mathbb{P}}(Y \mid X, A=a)\}} -\log g(Y^a; \beta_a) = -\int_{y \in \mathcal{Y}} \log g(y; \beta_a) \mathbb{P}_n\{\hat{\mathbb{P}}(Y = y \mid X, A = a)\}\, \mathrm{d}y
$$

$$
\approx \begin{cases} -h \sum_{j=1}^K \log g(y_j; \beta_a)\, \mathbb{P}_n\{\hat{\mathbb{P}}(Y = y_j \mid X, A = a)\}, & \text{if } d_Y = 1, \\ -\mathbb{P}_K\{\log g(Y^a; \beta_a)\}, & \text{if } d_Y > 1, \end{cases} \tag{9}
$$

where $y_{\min} \leq y_1 < \cdots < y_K \leq y_{\max}$ is an equidistant grid of points on $\mathcal{Y}$ with step size $h$, and $\{Y_j^a\}_{j=1}^K$ is an i.i.d. sample drawn from $\mathbb{P}_n\{\hat{\mathbb{P}}(Y \mid X, A = a)\}$.

**Training.** To train both components in our INFs, we make use of a two-step training procedure. Specifically, we first fit the nuisance parameters with the teacher flow. Then, we freeze the parameters of the teacher flow and fit the student flow. We additionally employ exponential moving average (EMA) of the student parameters with a smoothing hyperparameter $\gamma$ to stabilize the training for small minibatch sizes (Polyak & Juditsky, 1992). We show the full algorithm in Appendix D and further implementation details in Appendix E.

**Inference time.** One main advantage of our teacher-student model is that the student flow has constant inference time (e.g., during the evaluation phase). Hence, contrary to state-of-the-art baselines, the inference of our INFs do not depend on the dimensionality of covariates and the size of the training data. This is a major advantage over semi-parametric plug-in estimators. For a detailed runtime comparison, we refer to Appendix K. As such, the student flow allows our method to scale well to large datasets as in medicine (Johnson et al., 2016).

## 5.2 One-step bias correction

To provide an efficient estimation for the parameters of the student flow, we augment the cross-entropy loss (Eq. (9)) with a one-step bias correction. To evaluate the bias correction term, we need to compute an approximation of the conditional cross-entropy loss ($\mathcal{L}_{\mathrm{CCE}}(X; \beta_a)$) as in Eq. (9). We thus compute

$$
\mathcal{L}_{\mathrm{CCE}}(X; \beta_a) = \mathop{\mathbb{E}}_{Y \sim \hat{\mathbb{P}}(Y \mid X, A=a)} -\log g(Y; \beta_a) \approx \begin{cases} -h \sum_{j=1}^K \log g(y_j; \beta_a)\, \hat{\mathbb{P}}(Y = y_j \mid X, A = a), & \text{if } d_Y = 1, \\ -\mathbb{P}_K\{\log g(Y^{X,a}; \beta_a)\}, & \text{if } d_Y > 1, \end{cases}
$$

where $y_{\min} \leq y_1 < \cdots < y_K \leq y_{\max}$ is an equidistant grid of points on $\mathcal{Y}$ with step size $h$, and $\{Y_j^{X,a}\}_{j=1}^K$ is an i.i.d. sample drawn from $\hat{\mathbb{P}}(Y \mid X, A = a)$. Finally, we obtain the loss of the

---

[3]This is standard approach in neural conditional density estimation (Bishop, 1994; Kingma & Welling, 2013).

[4]One can use a single normalizing flow with a hypernetwork for categorical treatments.

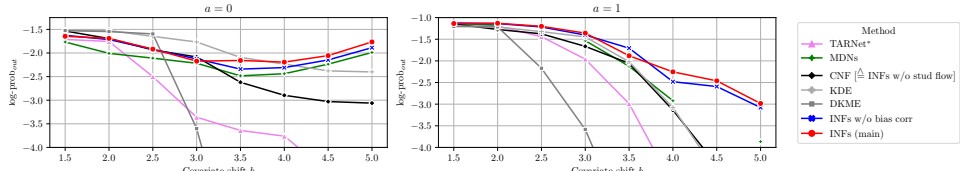

Figure 3: Results for synthetic data based on the SCM from Figure 1. Reported: mean over ten-fold train-test splits. Some runs for MDNs resulted with the log-prob$_{\text{out}}$ = $-\infty$ and, thus, are not shown.

student flow ($\mathcal{L}_s$), which is now suitable for our A-IPTW estimation from Eq. (9). We thus yield

$$\mathcal{L}_s(\beta_a) = \mathcal{L}_{\text{CE}}(\beta_a) + \mathbb{P}_n\left\{ \frac{\mathbb{1}(A = a)}{\hat{\pi}_a(X)}\Big( -\log g(Y; \beta_a) - \mathcal{L}_{\text{CCE}}(X; \beta_a)\Big)\right\}. \tag{10}$$

## 6 EXPERIMENTS

### 6.1 OVERVIEW

To show the effectiveness of our INFs, we use established (semi-)synthetic datasets that have been previously used for treatment effect estimation (Shi et al., 2019; Curth & van der Schaar, 2021). The benefit of (semi-)synthetic datasets is that both factual and counterfactual outcomes are available (i.e., $Y_i^{\text{f}}$ and $Y_i^{\text{cf}}$). Therefore, we can obtain a sample from the ground-truth interventional distribution, i.e., $Y[a]_i = \mathbb{1}(A_i = a)\,Y_i^{\text{f}} + \mathbb{1}(A_i \neq a)\,Y_i^{\text{cf}}$, which we can then use for IDE benchmarking.

**Evaluation metric.** We use the average log-probability as our standard metric for comparing density estimators. It is given by log-prob$_{\mathcal{D}} = \frac{1}{n}\sum_{i=1}^n \log \hat{\mathbb{P}}(Y[a] = Y[a]_i)$, where higher values indicate a better fit. The maximum value of the average log-probability is upper-bounded by the entropy, which, in general, is different for each potential outcome. Therefore, we separately report the results for each potential outcome. Of note, the log-probability is equivalent to the empirical KL-divergence.

**Baselines.** We use state-of-the-art IDE baselines (see Sec. 2.1): (1) an extended TARNet (**TARNet***) (Shalit et al., 2017) estimating the mean of a conditional homoscedastic normal distribution; (2) mixture density networks (**MDNs**) (Bishop, 1994)[5]; (3) conditional normalizing flow (**CNF**) (Trippe & Turner, 2018); (4) kernel density estimation (**KDE**) (Kim et al., 2018); and (5) distributional kernel mean embeddings (**DKME**) (Muandet et al., 2021). TARNet*, MDNs, and CNF are semi-parametric plug-in estimators (see Eq. (2)). Importantly, KDE and DKME do not guarantee a proper density estimation (unlike our INFs). We thus performed an additional re-normalization and negative values clipping, so that we can use the average log-probability as an evaluation metric. Details on the baselines are in Appendix F, and hyperparameter tuning is reported in Appendix G.

**Ablation studies.** We compare three variants of our INFs: (1) **INFs** (main): Our INFs as introduced above using A-IPTW estimation. (2) **INFs w/o stud flow**: A simplified variant which uses only the conditional density estimation from the teacher flow as a semi-parametric plug-in estimator, and thus without student flow. This variant is identical to the CNF baseline. (3) **INFs w/o bias corr**: We use the covariate-adjusted fully-parametric estimator, where the student flow only uses the cross-entropy loss from Eq. (9) but without one-step bias correction. The ablations have the same hyperparameters as our main method for better comparability.

### 6.2 RESULTS

**Synthetic data.** We generate synthetic data using the SCM ($d_X = 1$) from Fig. 1. Here, we vary the covariate shift $b$, which controls the overlap between the treated and non-treated population. Notably, low values of $b$ correspond to the case, where both populations are similar or the same, while high values of $b$ result in the violation of the positivity assumption. Further details on the synthetic dataset are provided in Appendix H. Fig. 3 shows the results. Our INFs achieve clear performance improvements over the baselines, especially for larger $b$. Moreover, the ablation studies confirm that our proposed deep learning architecture with one-step bias correction is superior. In Appendix I, we additionally provide a two-dimensional benchmark, where our INFs prove their effectiveness.

**IHDP dataset.** The Infant Health and Development Program (IHDP) (Hill, 2011) is a semi-synthetic dataset with two synthetic potential outcomes generated from real-world med-

---

[5]MDNs were previously used to estimate the conditional distribution of outcome for quantifying the ignorance regions of ITE estimation (Jesson et al., 2021; 2022). However, this is different from our IDE task.

ical covariates ($n = 747, d_X = 25$, see details in Appendix H). Here, we used ten-fold train/test splits (90%/10%) and perform hyperparameter tuning based on the first split. Results are in Table 2. TARNet* is known to entail a ground-truth conditional distribution model and should thus not be interpreted as a baseline but as an upper performance bound. Our INFs reach an equally good performance and, importantly, outperform all the other baselines for both potential outcomes. The ablation study again confirms that our main INFs are superior over the other variants without the student flow and without bias correction. In Appendix J, we repeat the evaluation using the empirical Wasserstein distance with similar findings.

Table 2: Results for IHDP dataset. Reported: mean $\pm$ sd.

| | $a = 0$ | | $a = 1$ | |
|---|---|---|---|---|
| | log-prob$_{\text{in}}$ | log-prob$_{\text{out}}$ | log-prob$_{\text{in}}$ | log-prob$_{\text{out}}$ |
| TARNet* [$\hat{=}$ ground-truth for IHDP] | $-0.919 \pm 0.011$ | $\mathbf{-0.928 \pm 0.088}$ | $\mathbf{-0.635 \pm 0.010}$ | $\mathbf{-0.634 \pm 0.075}$ |
| MDNs | $-0.927 \pm 0.024$ | $-0.942 \pm 0.080$ | $-0.679 \pm 0.048$ | $-0.684 \pm 0.077$ |
| CNF [$\hat{=}$ INFs w/o stud flow] | $-0.943 \pm 0.032$ | $-0.970 \pm 0.072$ | $-0.679 \pm 0.061$ | $-0.674 \pm 0.091$ |
| KDE (Kim et al., 2018) | $-0.942 \pm 0.010$ | $-0.948 \pm 0.069$ | $-0.700 \pm 0.044$ | $-0.708 \pm 0.098$ |
| DKME (Muandet et al., 2021) | $-0.940 \pm 0.010$ | $-0.952 \pm 0.082$ | $-0.665 \pm 0.015$ | $-0.670 \pm 0.063$ |
| INFs w/o bias corr | $-0.932 \pm 0.013$ | $-0.936 \pm 0.112$ | $-0.667 \pm 0.028$ | $-0.670 \pm 0.067$ |
| INFs (main) | $\mathbf{-0.912 \pm 0.010}$ | $\underline{-0.929 \pm 0.099}$ | $\underline{-0.658 \pm 0.020}$ | $\underline{-0.659 \pm 0.090}$ |

Higher = better (best in bold, second best underlined)

**ACIC 2016 & 2018 datasets.** ACIC 2016 & 2018 provide a collection of semi-synthetic datasets with various data-generating mechanisms (Dorie et al., 2019; Shimoni et al., 2018) (see details in Appendix H). We select 15 random datasets from ACIC 2016 and 24 random datasets (4 of each of 6 sizes) from ACIC 2018. We perform five random train/test splits (80%/20%) for each dataset, tune hyperparameters on the first split and evaluate the average in- and out-sample log-probability on every split. Table 3 provides the performance comparison. Again, our INFs have a clear performance improvement over both baselines and other model variants. Compared to MDNs as the second-best method, our INFs scale much better in terms of runtime, especially for large sample sizes (see Appendix K).

Table 3: Results for ACIC 2016 and ACIC 2018. Reported: % of runs with the best performance.

| | ACIC 2016 (77 datasets) | | ACIC 2018 (24 datasets) | |
|---|---|---|---|---|
| | % best$_{\text{in}}$ | % best$_{\text{out}}$ | % best$_{\text{in}}$ | % best$_{\text{out}}$ |
| TARNet* | 4.55% | 6.88% | 8.33% | 10.42% |
| MDNs | 30.26% | 31.43% | 21.67% | 19.17% |
| CNF [$\hat{=}$ INFs w/o stud flow] | 15.97% | 17.14% | 15.42% | 14.58% |
| KDE (Kim et al., 2018) | 1.04% | 1.17% | 10.42% | 9.58% |
| DKME (Muandet et al., 2021) | 0.39% | 0.78% | 8.75% | 10.83% |
| INFs w/o bias corr | 6.75% | 8.57% | 5.00% | 7.08% |
| INFs (main) | **41.04%** | **34.03%** | **30.42%** | **28.33%** |

Higher = better (best in bold)

**HC-MNIST dataset.** Hidden confounding MNIST dataset is a semi-synthetic dataset (Jesson et al., 2021), constructed on top of the canonical image dataset of handwritten digits (MNIST) (LeCun, 1998). To satisfy the exchangeability assumption, we add a hidden confounder to the set of all covariates, i. e., 28x28 images ($d_X = 784 + 1$). For dataset details, see Appendix H. For our experiments, we use only the train subset of the original MNIST ($n = 42,000$). Here we used ten random train/test splits (80%/20%) and tune hyperparameters on the first split. Table 4 shows the results of the experiments. Note that the non- and semi-parametric baselines suffer from scalability issues and were thus excluded. Further, our INFs outperform the variant without a bias correction, i. e., the only other available baseline.

Table 4: Results for HC-MNIST. Reported: mean $\pm$ sd. over ten random train-test splits.

| | $a = 0$ | | $a = 1$ | |
|---|---|---|---|---|
| | log-prob$_{\text{in}}$ | log-prob$_{\text{out}}$ | log-prob$_{\text{in}}$ | log-prob$_{\text{out}}$ |
| INFs w/o bias corr | $-1.43 \pm 0.18$ | $-1.43 \pm 0.18$ | $-1.40 \pm 0.17$ | $-1.40 \pm 0.17$ |
| INFs (main) | $\mathbf{-1.34 \pm 0.01}$ | $\mathbf{-1.34 \pm 0.01}$ | $\mathbf{-1.33 \pm 0.00}$ | $\mathbf{-1.33 \pm 0.01}$ |

Higher = better (best in bold)

**Scalability.** Experiments with ACIC 2018 and HC-MNIST datasets showed high effectiveness of our INFs for datasets with large sample sizes ($n > 25,000$) and with high-dimensional covariates ($d_X > 100$). We provide a runtime comparison in Appendix K. For HC-MNIST, non- and semi-parametric methods even become completely impractical due to memory and time constraints. Importantly, this is a major advantage of our fully-parametric IDE estimator, INFs, over semi-paramentric plug-in etimators and other baselines.

**Case study.** We performed a case study using data from California's tobacco control program to estimate its effect on tobacco sales. Previously, the evidence was primarily based on point estimates without information on the interventional density (Abadie et al., 2010). Our INFs suggest that the program led to a large reduction in tobacco sales .

**Discussion.** Interestingly, both components are important for the final performance (see our ablation studies). First, the teacher flow with the help of noise regularization performs consistent estimation of the nuisance parameters. Second, the student flow uses estimated nuisance parameters to solve the optimization objective. The student flow is *not* redundant but crucial for computational performance. While simple NFs have a similar estimation performance in terms of goodness-of-fit, only our INFs have constant inference time (e.g., during the evaluation phase regardless of the data size). This is a major advantage of parametric treatment effect estimators over semi-paramentric plug-in estimators.

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

## A  RELATED WORK: NORMALIZING FLOWS FOR CAUSAL INFERENCE

NFs have been used in the wider area of causal inference, yet in vastly *different* tasks than ours. Examples include, e. g., robust prediction by employing causal mechanisms (Müller et al., 2021); combining interventional and observational datasets (Ilse et al., 2021); and causal discovery (Brouillard et al., 2020). Further, several works aim to model Bayesian networks or structural causal models (SCMs) with known or unknown causal diagrams. For example, NFs were used as a probabilistic model for Bayesian networks aimed at causal discovery, as well as downstream interventional and counterfactual inference (Khemakhem et al., 2021; Wang et al., 2021; Wehenkel & Louppe, 2021). Balgi et al. (2022) build upon a temporal SCM with exogenous noise, where NFs are used for interventional and counterfactual queries. Importantly, all the aforementioned methods assume continuous variables in SCMs and *independence of exogenous noise*.[6] Hence, these methods are **not** applicable in our case, which considers semi-Markovian SCMs and which is thus a different inference task.[7] In sum, NFs have not yet been adapted to IDE, which is our novelty.

---

[6]This is commonly known as a causal Markov condition.

[7]This is stated in our identifiability assumptions: there is no limitation on the exogenous noise independence between outcome and covariates. Hence, our setting is more general.

# B  BACKGROUND MATERIALS

## B.1  NORMALIZING FLOWS

Normalizing flows (NFs) (Tabak & Vanden-Eijnden, 2010; Rezende & Mohamed, 2015) are flexible probabilistic models with a tractable density. A normalizing flow describes the change of the density of a continuous random variable after applying a sequence of invertible transformations. Given a random variable $Z$ with some known density $\mathbb{P}(Z = \cdot)$, e. g., normal or uniform, we define a transformed variable

$$X = t(Z) \quad Z \sim \mathbb{P}(Z), \tag{11}$$

where $t(\cdot) : \mathcal{Z} \to \mathcal{X}$ denotes an invertible forward transformation with inverse $t^{-1}(\cdot) : \mathcal{X} \to \mathcal{Z}$. Importantly, the transformation is defined between spaces of same dimensionality $d_Z = d_X$. To find a distribution of $X$, we can apply the multivariate *change of variables formula*

$$\mathbb{P}(X = x) = \mathbb{P}(Z = z) \left| \det \frac{\mathrm{d}Z}{\mathrm{d}X} \right| = \mathbb{P}(Z = t^{-1}(x)) \left| \det \frac{\mathrm{d}t^{-1}}{\mathrm{d}X}(x) \right|, \tag{12}$$

where $\det \frac{\mathrm{d}t^{-1}}{\mathrm{d}X}(x)$ is the Jacobian determinant of the inverse transformation $t^{-1}(\cdot)$. Then, using the *inverse function theorem*, we obtain

$$\frac{\mathrm{d}t^{-1}}{\mathrm{d}X} = \left( \frac{\mathrm{d}t}{\mathrm{d}Z} \right)^{-1}, \tag{13}$$

so that the Jacobian of the inverse transformation can be substituted with the inverse Jacobian of forward transformation. Using the properties of the determinant, Eq. (12) can be simplified to

$$\mathbb{P}(X = x) = \mathbb{P}(Z = t^{-1}(x)) \left| \det \frac{\mathrm{d}t}{\mathrm{d}Z}(t^{-1}(x)) \right|^{-1}. \tag{14}$$

The name *normalizing* comes from the fact that any regular continuous distribution $X$ can be transformed to a normal $Z$ with a specific $t^{-1}(\cdot)$.

We can construct arbitrarily complex densities by applying a composition of $K$ transformations $t_1, t_2, \ldots, t_K$:

$$X = Z_K = t_K(Z_{K-1}) = t_K(t_{K-1}(Z_{K-2})) = \ldots = t_K \circ \ldots \circ t_1(Z_0), \tag{15}$$

where $Z_0$ is called a base distribution. One calls this chain of transformations a *flow*. Finally, the density of $X$ can be recursively found as

$$\mathbb{P}(Z_K = z_K) = \mathbb{P}(Z_{K-1} = z_{K-1}) \left| \det \frac{\mathrm{d}t_K}{\mathrm{d}Z_{K-1}}(z_{K-1}) \right|^{-1} = \mathbb{P}(Z_0 = z_0) \prod_{k=1}^{K} \left| \det \frac{\mathrm{d}t_k}{\mathrm{d}Z_{k-1}}(z_{k-1}) \right|^{-1}, \tag{16}$$

where $z_0, z_1, \ldots, z_K$ are found via Eq. (15). Consequently, we now can directly evaluate the log-likelihood of an observation $X_i = Z_{Ki}$ and, with a proper parametrization of transformations, back-propagate trough it. Examples of simple transformations include affine, planar, and radial (Rezende & Mohamed, 2015).

## B.2  CAUSAL MODEL AND INDENTIFICATION

In this section, we provide a brief background on the underlying causal model in this paper, using both the potential outcomes and the structural causal model framework. These frameworks are equivalent in the sense that they both allow for identification of the interventional density and yield the same statistical estimand.

**Potential outcomes framework.** The observed variables in our model are covariates $X \in \mathcal{X} \subseteq \mathbb{R}^{d_X}$, a treatment $A \in \{0, 1\}$, and a $d_Y$-dimensional continuous outcome $Y \in \mathcal{Y} \subseteq \mathbb{R}^{d_Y}$. In the main paper, we used the potential outcomes framework (Rubin, 1974) to define the causal estimates. In particular, we defined $Y[a]$ as the potential outcome after intervening on treatment by setting it to $a$. By imposing Assumptions (1)–(3) in Section 3, this allows us to define the interventional density (our causal estimand) via

$$\mathbb{P}(Y[a] = y) = \int_{x \in \mathcal{X}} \mathbb{P}(Y = y \mid X = x, A = a) \mathbb{P}(X = x) \, \mathrm{d}x = \mathop{\mathbb{E}}_{X \sim \mathbb{P}(X)} \big( \mathbb{P}(Y = y \mid X, A = a) \big). \tag{17}$$

**SCM framework.** Equivalently to the potential outcomes framework, we can also define the interventional density within the structural causal model (SCM) framework (Pearl, 2009; Bareinboim et al., 2022). More precisely, we can define a (semi-Markovian) SCM by introducing independent exogenous latent variables $U_A \sim \mathbb{P}(U_A)$ and $U_{XY} \sim \mathbb{P}(U_{XY})$; and the functional assignments $X := f_X(U_{XY})$, $A := f_A(X, U_A)$, and $Y := f_Y(X, U_{XY})$. Here, $X, A$ and $Y$ are observed endogenous variables, satisfying Assumptions (1)–(3). We show a corresponding causal graph in Fig. 4.

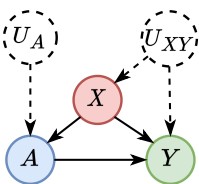

Figure 4: Causal graph corresponding to the potential outcome framework assumptions.

**Interventions vs. counterfactuals.** We follow Pearl's hierarchy on causal inference (Bareinboim et al., 2022) and distinguish the *interventional* and the *counterfactual* distribution. In SCM language, we can use Pearl's do-notation $do(A) = a$ to denote an intervention on the treatment $A$. This corresponds to setting $A = a$ in a graph $G_a$ where all arrows from parent nodes of $A$ to $A$ are removed.

We can then define the potential outcome $Y[a]$ via its interventional density

$$\mathbb{P}(Y[a] = y) = \mathbb{P}(Y = y \mid do(A = a)) \tag{18}$$

and obtain the identification result from Eq. (17).

In contrast, counterfactual queries aim to answer *individualized* questions "what would have happened if we had used a different treatment *given already treated or untreated population*". We can then define the *counterfactual density* as

$$\mathbb{P}(Y[a'] = y \mid A = a) \tag{19}$$

for some different treatment $a' \neq a$. This is the distributional equivalent to the average treatment effect of the treated (ATT). However, most of the treatment effect estimation literature focuses on *interventional* causal estimands (such as the ATE). Our paper is therefore in line with previous work. We acknowledge that other papers oftentimes call the interventional distribution counterfactual distributions for simplicity.

**Comparison to other identification strategies.** For the identification of the interventional density, we mainly rely on the three main assumptions positivity, consistency, and exchangeability (or, equivalently, on the back-door adjustment from Eq. (17)). This is a common setup in treatment effect estimation (van der Laan & Rubin, 2006; Shalit et al., 2017; Wager & Athey, 2018). More complex adjustment rules (e.g., front-door adjustment, adjustment for napkin graph) have the following limitations: (1) they require more unusual, complex assumptions which are often violated in practice; and (2) they require a complex efficient estimation theory Vowels et al. (2022). Nevertheless, this could be an interesting direction for future research.

### B.3 EFFICIENCY THEORY AND INFLUENCE FUNCTIONS

In this section, we give a brief background on semi-parametric efficiency theory and influence functions. Our background builds upon Kennedy et al. (2021), and we thus refer to it for mathematical details and further explanations.

Let us consider a semi-parametric statistical model $\{\mathbb{P} \in \mathcal{P}\}$, where $\mathcal{P}$ is a family of probability measures. We are interested in estimating a functional $\psi : \mathcal{P} \to \mathbb{R}$. If $\psi$ is sufficiently smooth, it admits the so-called *von Mises-* or *distributional Taylor expansion*

$$\psi(\bar{\mathbb{P}}) - \psi(\mathbb{P}) = \int \phi(t, \bar{\mathbb{P}}) \, \mathrm{d}(\bar{\mathbb{P}} - \mathbb{P})(t) + R_2(\bar{\mathbb{P}}, \mathbb{P}), \tag{20}$$

where $R_2(\bar{\mathbb{P}}, \mathbb{P})$ is a *second-order remainder term* and $\phi(t, \mathbb{P})$ is the so-called *efficient influence function* of $\psi$, satisfying $\int \phi(t, \mathbb{P}) d\mathbb{P}(t) = 0$ and $\int \phi(t, \mathbb{P})^2 d\mathbb{P}(t) < \infty$.

The efficient influence function $\phi(\cdot, \cdot)$ plays an important role in the theory of efficient semi-parametric estimation. Under certain assumptions, it can be shown that, for any sequence of estimators $\hat{\psi}_n$, it holds true that

$$\inf_{\delta > 0} \liminf_{n \to \infty} \sup_{\mathrm{TV}(\mathbb{P}, \mathbb{Q}) < \delta} n \, \mathbb{E}_{\mathbb{Q}} \left[ (\hat{\psi}_n - \psi(\mathbb{Q}))^2) \right] \geq \mathrm{var}\left(\phi(T, \mathbb{P})\right), \tag{21}$$

where TV denotes total variation. Hence, $\phi$ characterizes the best possible variance an estimator can achieve (in a local min-max sense).

Let now $\hat{\mathbb{P}}$ be an estimator of $\mathbb{P}$ and $\psi(\hat{\mathbb{P}})$ the so-called *plug-in estimator* of $\psi(P)$. The von Mises expansion from Eq. (20) implies that $\psi(\hat{\mathbb{P}})$ yields a first-order *plug-in bias* because

$$\psi(\hat{\mathbb{P}}) - \psi(\mathbb{P}) = - \int \phi(t, \hat{\mathbb{P}}) \, d\mathbb{P}(t) + R_2(\hat{\mathbb{P}}, \mathbb{P}) \tag{22}$$

due to that $\int \phi(t, \hat{\mathbb{P}}) \, d\hat{\mathbb{P}}(t) = 0$. A simple way to correct for the plug-in bias is to estimate the bias term from the right-hand side of Eq. (22) and add it to the plug-in estimator via

$$\hat{\psi}^{\text{A-IPTW}} = \psi(\hat{\mathbb{P}}) + \mathbb{P}_n(\phi(T, \hat{\mathbb{P}})). \tag{23}$$

Under certain assumptions it, can be shown that the bias-corrected estimator $\hat{\psi}^{\text{A-IPTW}}$ is asymptotically normal with mean zero and variance $\mathrm{var}\left(\phi(T, \mathbb{P})\right)$. Hence, by Eq. (21), $\hat{\psi}^{\text{A-IPTW}}$ is (asymptotically) *efficient* in the sense that it is consistent with the best possible variance.

**Application to interventional density estimation**: We now return to the specific statistical model in our paper, i.e., we aim at interventional density estimation. In other words, the estimand $\psi(\mathbb{P})$ we are interested in is the function

$$\mathbb{P}(Y[a] = \cdot) = \underset{X \sim \mathbb{P}(X)}{\mathbb{E}} \left( \mathbb{P}(Y = \cdot \mid X, A = a) \right). \tag{24}$$

Given an initial estimator $\hat{\mathbb{P}}(Y = \cdot \mid X, A = a)$ and the marginal empirical probability measure $\mathbb{P}_n\{\cdot\}$, the plug-in estimator becomes

$$\hat{\mathbb{P}}^{\text{PI}}(Y[a] = \cdot) = \mathbb{P}_n\{\hat{\mathbb{P}}(Y = \cdot \mid X, A = a)\}. \tag{25}$$

As described above, this estimator suffers from plug-in bias and is not efficient. However, a one-step bias correction for our setting is not as simple due to the fact that the interventional density is a functional target estimand and, hence, infinite dimensional. As a remedy, Kennedy et al. (2021) proposes an elegant solution by introducing the finite dimensional projection parameter

$$\hat{\beta}_a = \underset{\beta_a}{\arg \min} \, \mathrm{KL}\left( \mathbb{P}(Y[a]) \, \middle\| \, g(\cdot; \beta_a) \right), \tag{26}$$

which is equivalent to solving the moment condition

$$m(\beta_a) = \underset{X \sim \mathbb{P}(X)}{\mathbb{E}} \left( \mathbb{E}\left( T(Y; \beta_a) \mid X, A = a \right) \right) \overset{!}{=} 0, \tag{27}$$

where $T = T(Y; \beta_a) = -\nabla_{\beta_a} \log g(Y; \beta_a)$. The advantage of this approach is that the moment $m(\beta_a)$ is a finite dimensional quantity, which means efficiency theory can be applied. The plug-in estimator for the moment is

$$\hat{m}^{\text{PI}}(\beta_a) = \underset{Y^a \sim \mathbb{P}_n\{\hat{\mathbb{P}}(Y|X, A=a)\}}{\mathbb{E}} T(Y^a; \beta_a). \tag{28}$$

Kennedy et al. (2021) also derived the efficient influence function for the moment:

$$\phi_a(T; \mathbb{P}) = \frac{\mathbb{1}(A = a)}{\pi_a(X)} \left( T - \mathbb{E}(T \mid X, A = a) \right) + \mathbb{E}(T \mid X, A = a) - \underset{X \sim \mathbb{P}(X)}{\mathbb{E}} \left( \mathbb{E}(T \mid X, A = a) \right). \tag{29}$$

Hence, a bias-corrected estimator for the projection parameter can be obtained by solving

$$\hat{m}^{\text{A-IPTW}}(\beta_a) = \hat{m}^{\text{PI}}(\beta_a) + \mathbb{P}_n\{\phi_a(T(Y; \beta_a); \hat{\mathbb{P}})\} \overset{!}{=} 0. \tag{30}$$

Estimating the projection parameter via Eq. (30) requires solving a (potentially high-dimension) system of non-linear equations, which is often infeasible in practice. Hence, as a remedy, we propose in this paper to reformulate Eq. (30) as an optimization problem which can be incorporated directly into loss of a neural network (see Appendix C).

## C  BIAS-CORRECTED MOMENT CONDITION AS AN OPTIMIZATION TASK

We aim to transform the bias-corrected moment condition into an optimization objective:

$$\hat{m}^{\text{A-IPTW}}(\beta_a) = \hat{m}^{\text{PI}}(\beta_a) + \mathbb{P}_n\big\{\phi_a(T(Y;\beta_a);\hat{\mathbb{P}})\big\} \overset{!}{=} 0. \tag{31}$$

We first note that the plug-in estimator of moment condition $\hat{m}^{\text{PI}}(\beta_a)$ can be rewritten as

$$\hat{m}^{\text{PI}}(\beta_a) = \underset{Y^a \sim \mathbb{P}_n\{\hat{\mathbb{P}}(Y|X,A=a)\}}{\mathbb{E}} T(Y^a;\beta_a) = \int_{\mathcal{Y}} T(y;\beta_a)\,\mathbb{P}_n\{\hat{\mathbb{P}}(Y=y \mid X, A=a)\}\,\mathrm{d}y \tag{32}$$

$$= \mathbb{P}_n\Big\{ \int_{\mathcal{Y}} T(y;\beta_a)\,\hat{\mathbb{P}}(Y=y \mid X, A=a)\,\mathrm{d}y \Big\} = \mathbb{P}_n\Big\{ \hat{\mathbb{E}}\big(T(Y;\beta_a) \mid X, A=a\big)\Big\}, \tag{33}$$

where the last equality follows from the definition of the conditional expectation. Notably, we see that the moment condition could be equivalently solved with either the conditional distribution, $\mathbb{P}(Y \mid X, A=a)$, or with the functional regression, $\mathbb{E}\big(T(Y;\beta_a) \mid X, A=a\big)$.

Let us unroll the bias correction term of Eq. (7):

$$\mathbb{P}_n\{\phi_a(T;\hat{\mathbb{P}})\} = \mathbb{P}_n\bigg\{ \frac{\mathbb{1}(A=a)}{\hat{\pi}_a(X)}\Big(T - \hat{\mathbb{E}}(T \mid X, A=a)\Big) + \hat{\mathbb{E}}(T \mid X, A=a) - \mathbb{P}_n\{\hat{\mathbb{E}}(T \mid X, A=a))\} \bigg\} \tag{34}$$

$$= \mathbb{P}_n\bigg\{ \frac{\mathbb{1}(A=a)}{\hat{\pi}_a(X)}\Big(T - \hat{\mathbb{E}}(T \mid X, A=a)\Big) + \hat{\mathbb{E}}(T \mid X, A=a) \bigg\} - \mathbb{P}_n\{\hat{\mathbb{E}}(T \mid X, A=a))\}, \tag{35}$$

where nuisance parameters are marked with red color. Here, the last term is in fact the plug-in estimator of the moment condition, i.e., $-\hat{m}^{\text{PI}}(\beta_a)$. Therefore, we can simplify one-step bias corrected moment condition via

$$\hat{m}^{\text{A-IPTW}}(\beta_a) = \mathbb{P}_n\bigg\{ \frac{\mathbb{1}(A=a)}{\hat{\pi}_a(X)}\Big(T - \hat{\mathbb{E}}(T \mid X, A=a)\Big) + \hat{\mathbb{E}}(T \mid X, A=a) \bigg\} \tag{36}$$

$$= \underset{Y^a \sim \mathbb{P}_n\{\hat{\mathbb{P}}(Y|X,A=a)\}}{\mathbb{E}} T(Y^a;\beta_a) + \mathbb{P}_n\bigg\{ \frac{\mathbb{1}(A=a)}{\hat{\pi}_a(X)}\Big(T(Y;\beta_a) - \underset{Y \sim \hat{\mathbb{P}}(Y|X,A=a)}{\mathbb{E}} T(Y;\beta_a)\Big)\bigg\}, \tag{37}$$

where we use the conditional density estimator but not an estimator for the functional regression. This allows us to transform the A-IPTW moment condition into an optimization objective (Eq. (8)) by taking antiderivative with respect to $\beta_a$.

## D  TWO-STEP TRAINING PROCEDURE

Our INFs are trained with a two-step procedure. The procedure is shown in Algorithm 1. Recall that we use noise regularization as the main regularization technique for the teacher flow, and exponential moving average (EMA) for the student flow to stabilize training. A-IPTW estimation is also known to become unstable in a finite sample setting (Shi et al., 2019), so that inverse values of propensity score become too large. Thus, we manually discard observations with too small propensity score ($\hat{\pi}_a(X) < 0.05$) from bias correction.

---

**Algorithm 1** Training procedure of INFs

---

**Input:** number of iterations $n_{\text{iter},t}, n_{\text{iter},s}$; minibatch sizes $b_t, b_s$; learning rates $\eta_t, \eta_s$; intensities of the noise regularization $\sigma_x^2, \sigma_y^2$; EMA smoothing $\gamma$; grid size $K$.

**Init:** parameters of the teacher flow: $\text{FC}_1^{(0)}, \text{FC}_2^{(0)}$         ▷ Fitting the teacher flow
**for** i = 0 to $n_{\text{iter, t}}$ **do**
    $\mathcal{B} = \{X, A, Y\} \leftarrow$ minibatch of size $b_t$
    $R, \hat{\pi}_a(X) \leftarrow \text{FC}_1^{(i)}(X)$
    $\xi_x \sim N(0, \sigma_x^2); \xi_y \sim N(0, \sigma_y^2); \tilde{R} \leftarrow R + \xi_x; \tilde{Y} \leftarrow Y + \xi_y$     ▷ Noise regularization
    $\theta(X, A) \leftarrow \text{FC}_2^{(i)}(A, \tilde{R})$
    $\hat{\mathbb{P}}(Y \mid X, A) \leftarrow$ normalizing flow with parameters $\theta(X, A)$
    $\mathcal{L}_{\text{NLL}} \leftarrow -\log \hat{\mathbb{P}}(Y = \tilde{Y} \mid X, A)$
    $\mathcal{L}_\pi \leftarrow \text{BCE}(\hat{\pi}_A(X), A)$
    $\mathcal{L}_t(\hat{\mathbb{P}}, \hat{\pi}_a) \leftarrow \mathbb{P}_{b_t}^{\mathcal{B}}\{\mathcal{L}_{\text{NLL}} + \alpha \mathcal{L}_\pi\}$
    $\text{FC}_1^{(i+1)}, \text{FC}_2^{(i+1)} \leftarrow$ optimization step wrt. $\mathcal{L}_t(\hat{\mathbb{P}}, \hat{\pi}_a)$ with learning rate $\eta_t$
**end for**
**Output:** nuisance parameters: $\hat{\mathbb{P}}(Y \mid X, A), \hat{\pi}_a(X)$

**Init:** parameters of the student flows: $\beta_a^{(0)}, \beta_{a,\text{EMA}}^{(0)} \leftarrow \beta_a^{(0)}$     ▷ Fitting the student flows
**for** i = 0 to $n_{\text{iter, s}}$ **do**
    $\mathcal{B} = \{X, A, Y\} \leftarrow$ minibatch of size $b_s$
    **for** $a \in \{0, 1\}$ **do**
        $\mathcal{L}_{\text{CE}}(\beta_a^{(i)}) \leftarrow -h \sum_{j=1}^K \log g(y_j; \beta_a^{(i)}) \mathbb{P}_{b_s}^{\mathcal{B}}\{\hat{\mathbb{P}}(Y = y_j \mid X, A = a)\}$
        $\mathcal{L}_{\text{CCE}}(X; \beta_a^{(i)}) \leftarrow -h \sum_{j=1}^K \log g(y_j; \beta_a^{(i)}) \hat{\mathbb{P}}(Y = y_j \mid X, A = a)$
        $\text{bias correction}(\beta_a^{(i)}) \leftarrow \mathbb{P}_{b_s}^{\mathcal{B}}\left\{ \frac{\mathbb{1}(A=a \& \hat{\pi}_a(X) \geq 0.05)}{\hat{\pi}_a(X)}\left( -\log g(Y; \beta_a^{(i)}) - \mathcal{L}_{\text{CCE}}(X; \beta_a^{(i)}) \right) \right\}$
        $\mathcal{L}_s(\beta_a^{(i)}) \leftarrow \mathcal{L}_{\text{CE}}(\beta_a^{(i)}) + \text{bias correction}(\beta_a^{(i)})$
        $\beta_a^{(i+1)} \leftarrow$ optimization step wrt. $\mathcal{L}_s(\beta_a^{(i)})$ with learning rate $\eta_s$
        $\beta_{a,\text{EMA}}^{(i+1)} \leftarrow \gamma \beta_{a,\text{EMA}}^{(i)} + (1 - \gamma)\beta_a^{(i+1)}$     ▷ EMA update
    **end for**
**end for**
**Output:** $\hat{\beta}_a^{\text{A-IPTW}} \leftarrow \beta_{a,\text{EMA}}^{(n_{\text{iter, s}})}$

---

# E  INFS IMPLEMENTATION DETAILS

**Implementation.** We implemented our INFs using PyTorch and Pyro. For both teacher and student flow, we employ neural spline flows (Durkan et al., 2019) with standard normal ($N(0; 1)$) as a base distribution. Neural spline flows construct an invertible transformation of the base distribution with the help of monotonic rational-quadratic splines. They are characterized by two main hyperparameters: a number of knots $n_{\mathrm{knots}}$ and a span of the transformation interval $[-B; B]$. $n_{\mathrm{knots}}$ controls the smoothness of estimated density and $B$ affects the support of the transformation. In our experiments, we heuristically set $B = y_{\max} - y_{\min} + 5$.

For the teacher flow, we use fully-connected subnetworks each with one hidden layer (with $h = 10$ hidden units), and the dimensionality of representation is set to $r = 10$.

**Training.** During training (see full algorithm in Appendix D), we adopt noise regularization (Rothfuss et al., 2019) and add an independent Gaussian noise $\xi_x \sim N(0, \sigma_x^2), \xi_y \sim N(0, \sigma_y^2)$ to the representation and output of the teacher flow, i. e., $\tilde{R} = R + \xi_x; \tilde{Y} = Y + \xi_y$. For faster learning, we approximate a full-sample average $\mathbb{P}_n\{\cdot\}$ with a minibatch average $\mathbb{P}_b^{\mathcal{B}}\{\cdot\}$ for all the losses, where $b$ is the minibatch size. We use stochastic gradient descent (SGD) for fitting the parameters of the teacher flow, and Adam optimizer (Kingma & Ba, 2015) for the student flow with learning rates $\eta_t$ and $\eta_s$, respectively. We fix the weighting hyperparameters of the loss to $\alpha = 1$ and the EMA smoothing hyperparameter to $\gamma = 0.995$. The grid size for approximating the cross-entropy loss is set to $K = 100$. Both $y_{\min}$ and $y_{\max}$ are set to the empirical minimum and maximum of the train sub-sample.

Note that we would need sample splitting for training both flows to guarantee the asymptotic properties, i. e., efficiency and double robustness (see Kennedy et al., 2021, Remark 5). Nevertheless, we used all data for the both components and trained our INFs with an auxiliary regularization because sample splitting can affect the performance in settings with limited data. This is consistent with previous work on deep learning for efficient treatment effect estimation (Curth & van der Schaar, 2021).

**Hyperparameter tuning.** We perform extensive hyperparameter tuning only for the teacher flow. Hyperparameters for tuning include, e. g., number of knots of neural spline flows $n_{\mathrm{knots},t}$, the minibatch size $b_t$, the learning rate $\eta_t$, and the intensities of the noise regularization $\sigma_x^2, \sigma_y^2$. On the other hand, we discovered, that the student flow works well with the same plain set of hyperparameters in almost all the experiments. Those include the minibatch size $b_s = 64$ and the learning rate $\eta_t = 0.005$. The number of knots $n_{\mathrm{knots},s}$ is chosen at hand for each dataset. Further details on hyperparameter tuning are provided in Appendix G.

# F   BASELINES

In the following, we describe the baseline methods in detail. These are two naïve semi-parametric plug-in estimators: mixture density networks (**MDNs**) (Bishop, 1994) and conditional normalizing flow (**CNF**) (Trippe & Turner, 2018). Further, we use two state-of-the-art IDE baselines: kernel density estimation (**KDE**) (Kim et al., 2018) and distributional kernel mean embeddings (**DKME**) (Muandet et al., 2021).

## F.1   NAÏVE SEMI-PARAMETRIC PLUG-IN ESTIMATORS

Semi-parametric plug-in estimators estimate the conditional outcome distribution and perform averaging over covariates during evaluation, as introduced in Eq. (2).

TARNet*, MDNs, and CNF make use of hypernetworks (Ha et al., 2017), which take covariates $X$ and treatment $A$ as an input and output parameters, i. e., $\theta(X, A)$ of the estimated conditional distribution $\hat{\mathbb{P}}(Y \mid X, A)$. Hypernetwork architectures are considered to be state-of-the-art for neural conditional density estimation and can be found in, e. g., Gaussian mixtures (Bishop, 1994), variational autoencoders (Kingma & Welling, 2013), and normalizing flows (Trippe & Turner, 2018). For comparability, we use the same network structure of the teacher flow in our INFs as the hypernetwork for the conditional distribution parameters. This gives two fully-connected subnetworks stacked on each other, i. e. $FC_1$ and $FC_2$, as introduced in Section 5.1. To regularize both conditional distribution estimators, we use noise regularization (Rothfuss et al., 2019).

**TARNet**\*. The treatment-agnostic representation network (TARNet) (Shalit et al., 2017) was proposed to estimate nuisance parameters for ITE, i. e., conditional means of outcomes. To obtain density estimates as outputs, we report results from an extended variant which we refer to as to TARNet\*. Specifically, we extended the original TARNet by modeling conditional outcome distribution as a homoscedastic normal distribution. For this, we add one unconditional parameter of standard deviation, $\sigma$, so that the conditional density equals to

$$\hat{\mathbb{P}}(Y = y \mid X, A) = N(y; \mu(X, A), \sigma^2), \tag{38}$$

where $N(y; \mu, \sigma^2)$ is a density of the normal distribution, and $\mu(X, A)$ is conditional mean of outcome. Notably, we do not use the two separate outcome heads (as in original TARNet) but only one, i. e., $FC_2$. This is crucial to ensure a fair comparison with other plug-in estimators. We estimate the standard deviation $\sigma$ using maximum-likelihood. Note also that TARNet\* is restricted to normal conditional outcome distributions and thus is not a universal density estimator. In contrast to our INFs, TARNet\* is unable to capture heavy-tailed, multi-modal, and skewed distributions.

**MDNs.** Mixture density networks (Bishop, 1994) are built on top of mixture of normal distributions, and can approximate any density arbitrarily well (Titterington et al., 1985), i.e.,

$$\hat{\mathbb{P}}(Y = y \mid X, A) = \sum_{j=1}^{n_C} w_j(X, A) \, N(y; \mu_j(X, A), \sigma_j^2(X, A)) \tag{39}$$

where $n_C$ is a number of mixture components, $w_j \geq 0, \sum_{j=1}^{n_C} w_j = 1$ are mixture weights, and $N(y; \mu_j, \sigma_j^2)$ is a density of the normal distribution. In the case of MDNs, the hypernetwork outputs logits of mixture weights and parameters of the normal distribution (i.e., mean and logarithm of the standard deviation), i. e., $\theta = \{\text{logits}(w_j), \mu_j, \log \sigma_j\}$. Here, the number of mixture components $n_C$ controls the smoothness of the estimator and represents the main hyperparameter for tuning.

**CNF.** We implement conditional normalizing flow (Trippe & Turner, 2018) with the help of neural spline flows (Durkan et al., 2019). Neural spline flows construct an invertible function parameterized by $\theta$, i. e., $f(\cdot; \theta) : \mathbb{R} \to \mathbb{R}$, which is a monotonic rational-quadratic spline with $n_{\text{knots}}$ knots. This spline transforms the density of a base distribution on the interval $[-B; B]$. Outside of the interval, $f(\cdot)$ equals to the identity function. This allows us to perform flexible parametric density estimation with the help of the change of variables formula, i.e.,

$$\hat{\mathbb{P}}(Y = y \mid X, A) = N\left(f^{-1}(y; \theta(X, A)); 0, 1\right) \left| \frac{\mathrm{d}f}{\mathrm{d}Y}\left(f^{-1}(y; \theta(X, A))\right) \right|^{-1} \tag{40}$$

where $f^{-1}(\cdot; \theta)$ is the inverse transformation, and the density of standard normal distribution $N(y; 0, 1)$ serves as a base distribution. As already discussed in Appendix E, $B$ affects the support of transformation, and the number of knots $n_{\text{knots}}$ controls the smoothness of the estimator and represents the main hyperparameter for tuning.

### F.2 KERNEL DENSITY ESTIMATION (KDE)

Kernel density estimation (KDE) is a semi-parametric method for IDE (Kim et al., 2018). It builds upon the idea of a density functional, namely $T_y(Y; h_a)$, to transform a random variable $Y$ into a proper density via

$$T_y(Y; h_a) = \frac{1}{h_a} K\left(\frac{\|Y - y\|_2}{h_a}\right) = \frac{1}{h_a \sqrt{2\pi}} \exp\left(-\frac{\|Y - y\|_2^2}{2h_a^2}\right), \qquad (41)$$

where $K(x) = \frac{1}{\sqrt{2\pi}} \exp(-x^2/2)$ is a radial basis function (RBF) with a treatment-specific smoothing parameter $h_a$ called bandwidth, and $\|\cdot\|_2$ is the $L_2$-norm.

Robins & Rotnitzky (2001) proposed a semi-parametric plug-in estimator of interventional density

$$\hat{\mathbb{P}}^{\text{PI}}(Y[a] = y) = \mathbb{P}_n\Big\{\hat{\mathbb{E}}\big(T_y(Y; h_a) \mid X, A = a\big)\Big\}, \qquad (42)$$

where $\hat{\mu}_{a,y}(X) = \hat{\mathbb{E}}\big(T_y(Y; h_a) \mid X, A = a\big)$ is a functional regression of $X$ and $A$ on $T_y(Y; h_a)$. Kim et al. (2018) further extended this estimator to an efficient, A-IPTW-style semi-parametric estimator

$$\hat{\mathbb{P}}^{\text{A-IPTW}}(Y[a] = y) = \mathbb{P}_n\left\{\frac{\mathbb{1}(A = a)}{\hat{\pi}_a(X)}\Big(T_y(Y; h_a) - \hat{\mu}_{a,y}(X)\Big) + \hat{\mu}_{a,y}(X)\right\}, \qquad (43)$$

where $\hat{\pi}_a(X)$ is an estimator of the propensity score.

The main challenge here is building a functional regression $\hat{\mu}_{a,y}(X)$. Unfortunately, the work by Kim et al. (2018) does not provide effective, practical solutions. Even more so, Eq. (43) does not guarantee that the estimated density is proper, i.e., integrates to 1 and is positive, especially in a small sample regime or when the propensity score has extremely low values.

To estimate the nuisance parameters, namely, the propensity score and the functional regression, we use the same network structure as for the teacher flow of our INFs (see Section 5.1). In this way, we estimate the propensity score and perform a functional regression with two joined, fully-connected subnetworks (i.e., $\text{FC}_1$ and $\text{FC}_2$). The first subnetwork, $\text{FC}_1$, outputs a representation $R$ and estimates the propensity score. The second subnetwork, $\text{FC}_2$, then takes the representation $R$ and the treatment $A$, and performs an outcome regression: $\hat{Y} = \hat{\mathbb{E}}(Y \mid X, A)$. The functional expression, i. e., Eq. (41), is predicted via $\hat{\mu}_{a,y}(X) = T_y(\hat{\mathbb{E}}(Y \mid X, A); h_a)$. Although, this is a biased estimator of $\mu_{a,y}(X)$, it ensures a proper normalization, i.e., $\int_{\mathcal{Y}} \hat{\mu}_{a,y}(X)\, \mathrm{d}y = 1$.

To fit $\text{FC}_1$ and $\text{FC}_2$, we use the sum of mean-squared error ($\mathcal{L}_{\text{MSE}}$) and binary cross-entropy ($\mathcal{L}_\pi$) losses via

$$\mathcal{L}_{\text{KDE}}(\hat{\mathbb{E}}, \hat{\pi}_a) = \mathbb{P}_n\{\mathcal{L}_{\text{MSE}} + \alpha \mathcal{L}_\pi\} \text{ with } \mathcal{L}_{\text{MSE}} = (\hat{Y} - Y)^2; \quad \mathcal{L}_\pi = \text{BCE}(\hat{\pi}_A(X), A), \qquad (44)$$

where $\alpha$ is a hyperparameter. In our experiments, we set $\alpha = 1$ and fit the nuisance parameters (i.e., $\hat{\pi}_a$ and $\hat{\mathbb{E}}(Y \mid X, A)$) using the Adam optimizer with $n_{\text{iter}} = 10000$ iterations. Both learning rate $\eta$ and minibatch size $b$ are subject to hyperparameter tuning.

We employ a median heuristic (Garreau et al., 2017) for choosing the bandwidth $h_a$, i.e.,

$$h_a^{\text{med}} = \sqrt{\frac{1}{2} \text{Median}\big(\|Y_i - Y_j\|_2^2 \mid A = a\big)}, \quad 1 \le i < j \le n, \qquad (45)$$

where $\|\cdot\|_2$ is the $L_2$-norm, and where $Y_i, Y_j$ are observations from the train subset, conditioned on $A = a$. To address the numeric instability of the A-IPTW estimator, we discard observations with too small propensity scores ($\hat{\pi}_a(X) < 0.05$) from averaging in Eq. (43), similarly to our INFs.

### F.3 DKME

Distributional kernel mean embeddings (DKME) (Muandet et al., 2021) is a non-parametric plug-in estimator of interventional densities. This methods builds a kernel mean embedding (KME), namely, $\mu_{Y|X,A=a}$, for the conditional distribution $\mathbb{P}(Y \mid X, A = a)$ via

$$\mu_{Y|X,A=a}(y) := \mathbb{E}_{Y \sim \mathbb{P}(Y|X,A=a)} l_a(y, Y), \tag{46}$$

where $l_a(\cdot, \cdot)$ is a measurable positive definite kernel associated with a reproducing kernel Hilbert space $\mathcal{H}$, so that $\mu_{Y|X,A=a}$ provides a mapping from the space of conditional distributions to the space of functions $\mathcal{H}$. If $l_a(\cdot, \cdot)$ is properly normalized, then $\mu_{Y|X,A=a}(y)$ is in fact a conditional density estimator.

To estimate the KME of the conditional outcome distribution (conditional mean embedding), we use the i.i.d. sample $\mathcal{D} = \{X_i, A_i, Y_i\}_{i=1}^n$, and split it into control and treated subsamples: $\mathcal{D} = \{X_i^0, Y_i^0\}_{i=1}^{n_0} \cup \{X_i^1, Y_i^1\}_{i=1}^{n_1}$. Then, $\mu_{Y|X,A=a}$ can be estimated via

$$\hat{\mu}_{Y|X,A=a}(y) = \sum_{i=1}^{n_a} w_i^a(X) \, l_a(y, Y_i^a), \tag{47}$$

$$\left(w_1^a(X), \ldots, w_{n_a}^a(X)\right)^\mathsf{T} = (\mathbf{K}^a + n_a \varepsilon \mathbf{I})^{-1} \, \mathbf{k}^a(X) \in \mathbb{R}^{n_a}, \tag{48}$$

$$\mathbf{k}^a(X) = \left(k(X, X_1^a), \ldots, k(X, X_{n_a}^a)\right)^\mathsf{T} \in \mathbb{R}^{n_a}, \tag{49}$$

where $\mathbf{I} \in \mathbb{R}^{n_a \times n_a}$ is an identity matrix, $\varepsilon > 0$ is a regularization hyperparameter, $\mathbf{K}^a \in \mathbb{R}^{n_a \times n_a}$ is a kernel matrix with elements $\mathbf{K}_{ij}^a = k(X_i^a, X_j^a)$, and $k(\cdot, \cdot)$ is a second kernel representing conditional dependencies between $X$ and $Y$ (Grünewälder et al., 2012).

Muandet et al. (2021) further developed a KME for interventional distribution, i. e., $\mu_{Y[a]}$, and its empirical estimate, $\hat{\mu}_{Y[a]}$:

$$\mu_{Y[a]}(y) = \mathbb{E}_{X \sim \mathbb{P}(X)} \mu_{Y|X,A=a}(y) \tag{50}$$

$$\hat{\mu}_{Y[a]}(y) = \mathbb{P}_n\{\hat{\mu}_{Y|X,A=a}(y)\} = \sum_{i=1}^{n_a} \beta_i^a \, l_a(y, Y_i^a), \tag{51}$$

$$(\beta_1^a, \ldots, \beta_{n_a}^a)^\mathsf{T} = (\mathbf{K}^a + n_a \varepsilon \mathbf{I})^{-1} \, \tilde{\mathbf{K}}^a \, \mathbf{1}_m \in \mathbb{R}^{n_a}, \tag{52}$$

where $\tilde{\mathbf{K}}^a \in \mathbb{R}^{n_a \times n}$ is a kernel matrix with elements $\tilde{\mathbf{K}}_{ij}^a = k(X_i^a, X_j)$, and $\mathbf{1}_m = (1/n, \ldots, 1/n)^\mathsf{T}$.

For our experiments, we choose both kernels, i. e., outcome kernel, $l_a(\cdot, \cdot)$, and conditional kernel, $k(\cdot, \cdot)$, to be RBF kernels with bandwidth parameters $h_{a,l}$ and $h_k$, respectively. Therefore, $\hat{\mu}_{Y[a]}(y)$ represents a valid interventional density estimator. Nevertheless, due to small sample sizes, some $\beta_i^a$ could be negative and the estimated density ends up having negative values.

We set the bandwidth of the outcome kernel, $h_{a,l}$ according to the median heuristic from Eq. (45). The bandwidth of the conditional kernel $h_k$ and the regularization hyperparameter $\varepsilon$ are subjects to the hyperparameter tuning. Motivated by the interpretation of conditional mean embedding as kernel ridge regression (Grünewälder et al., 2012), we use out-sample MSE of the ridge regression with parameters $h_k$ and $\varepsilon$ as a tuning criterion.

# G  HYPERPARAMETER TUNING

We performed hyperparameters tuning for all the baselines based on five-fold cross-validation using the train subset. For each baseline, we performed a grid search with respect to different tuning criteria, evaluated on the validation subsets. Table 5 shows grids for hyperparameter tuning and other parameters, such as tuning criteria, number of training iterations, and optimizers. We aimed for a fair comparison and thus kept the number of parameters, network structures, and grid size similar across models. For the sake of reproducibility, we make the chosen hyperparameters for all the experiments public (see YAML files in our GitHub[8]).

Importantly, to facilitate the convergence of baseline methods, we additionally perform a standard normalization of both factual and counterfactual outcomes for all the datasets.

Table 5: Hyperparameter tuning for baselines.

| Model | Sub-model | Hyperparameter | Range / Value |
|---|---|---|---|
| TARNet* | — | Intensity of noise regularization ($\sigma_x^2$) | 0.0, 0.01, 0.05, 0.1 |
| | | Intensity of noise regularization ($\sigma_y^2$) | 0.0, 0.01, 0.05, 0.1 |
| | | Learning rate ($\eta$) | 0.001, 0.005 |
| | | Minibatch size ($b$) | 32, 64 |
| | | Tuning strategy | random grid search with 50 runs |
| | | Tuning criterion | $\mathcal{L}_{\text{NLL}}$ |
| | | Number of train iterations ($n_{\text{iter}}$) | 5000 |
| | | Optimizer | SGD (momentum = 0.9) |
| MDNs | — | Number of mixture components ($n_C$) | 5, 10, 20 |
| | | Intensity of noise regularization ($\sigma_x^2$) | 0.0, 0.01, 0.05, 0.1 |
| | | Intensity of noise regularization ($\sigma_y^2$) | 0.0, 0.01, 0.05, 0.1 |
| | | Learning rate ($\eta$) | 0.001, 0.005 |
| | | Minibatch size ($b$) | 32, 64 |
| | | Tuning strategy | random grid search with 50 runs |
| | | Tuning criterion | $\mathcal{L}_{\text{NLL}}$ |
| | | Number of train iterations ($n_{\text{iter}}$) | 5000 |
| | | Optimizer | SGD (momentum = 0.9) |
| KDE | — | Learning rate ($\eta$) | 0.001, 0.005, 0.1 |
| | | Minibatch size ($b$) | 32, 64, 128 |
| | | Tuning strategy | full grid search |
| | | Tuning criterion | $\mathcal{L}_{\text{MSE}} + \alpha \mathcal{L}_\pi$ |
| | | Number of train iterations ($n_{\text{iter}}$) | 10000 |
| | | Optimizer | Adam (betas=(0.9, 0.999)) |
| DKME | — | Kernel smoothness ($\sigma_k = 2h_k^2$) | 0.0001, 0.001, 0.01, 0.1, 1, 10, 20 |
| | | Regularization parameter ($\varepsilon$) | 0.0001, 0.001, 0.01, 0.1, 1, 10 |
| | | Tuning strategy | full grid search |
| | | Tuning criterion | MSE of ridge regression |
| INFs | Teacher flow | Number of knots ($n_{\text{knots},t}$) | 5, 10, 20 |
| | | Intensity of noise regularization ($\sigma_x^2$) | 0.0, 0.01, 0.05, 0.1 |
| | | Intensity of noise regularization ($\sigma_y^2$) | 0.0, 0.01, 0.05, 0.1 |
| | | Learning rate ($\eta_t$) | 0.001, 0.005 |
| | | Minibatch size ($b_t$) | 32, 64 |
| | | Tuning strategy | random grid search with 50 runs |
| | | Tuning criterion | $\mathcal{L}_{\text{NLL}}$ |
| | | Number of train iterations ($n_{\text{iter},t}$) | 5000 |
| | | Optimizer | SGD (momentum = 0.9) |
| | Student flow | Number of knots ($n_{\text{knots},s}$) | dataset specific* |
| | | Learning rate ($\eta_s$) | 0.005 |
| | | Minibatch size ($b_s$) | 64 |
| | | Tuning strategy | w/o tuning |
| | | Number of train iterations ($n_{\text{iter},s}$) | 4000 |
| | | Optimizer | Adam (betas=(0.9, 0.999)) |

* $n_{\text{knots},s} = 5$ (synthetic data), $= 10$ (IHDP, HC-MNIST datasets), $= n_{\text{knots},t}$ (ACIC 2016 & 2018 datasets)

---

[8]https://anonymous.4open.science/r/AnonymousInterFlow-E2F3

# H DATASET DETAILS

## H.1 SYNTHETIC DATASET

We sample $n = 1000$ observations from the SCM (Fig. 1) and use a ten-fold split for train/test samples (90%/10%). We separately perform hyperparameter tuning based on the first split for each baseline and each level $b$. We then report an average out-sample log-likelihood over ten folds.

## H.2 IHDP DATASET

The IHDP dataset (Hill, 2011) uses a real-world dataset with 25 covariates (6 continuous, 19 binary) and one binary treatment, capturing aspects related to children and their mothers. Both treated and untreated, synthetic outcomes of IHDP are sampled from different conditional normal distributions. These distributions are homoscedastic ($\sigma^2 = 1$) but have substantially different conditional means. We used the setting "B" in (Hill, 2011) with a following SCM:

$$\begin{cases} X \sim \text{Real-World}(\cdot), \\ A \sim \text{Real-World}(X), \\ Y \sim N\big(A\,(X\beta - \omega) + (1 - A)\,(\exp((X + W)\beta)); 1\big), \end{cases} \tag{53}$$

where $\beta, W, \omega$ are constant parameters of the simulation. For the further details, we refer to (Hill, 2011).

## H.3 ACIC 2016 & 2018 DATASETS

Covariates of ACIC 2016 are taken from a large study of developmental disorders (Niswander, 1972), and covariates of ACIC 2018 are derived from the linked birth and infant death data (MacDorman & Atkinson, 1998). ACIC 2016 and ACIC 2018 differ in the number of true confounders, the varying level of overlap, and the form of conditional outcome distributions. ACIC 2016 has 77 different data-generating mechanisms with 100 equal-sized samples for each mechanism ($n = 4802, d_X = 82$).[9] ACIC 2018 provides 63 distinct data-generating mechanisms with around 40 non-equal-sized samples for each mechanism ($n$ ranges from $1,000$ to $50,000$, $d_X = 177$). Notably, ACIC 2018 has a constant ITE for most of the datasets, but heterogeneous propensity scores.

## H.4 HC-MNIST

Jesson et al. (2021) introduced a complex high-dimensional, semi-synthetic dataset based on the MNIST image dataset LeCun (1998), namely HC-MNIST. This dataset maps high-dimensional images onto a one-dimensional manifold, where potential outcomes depend in a complex way on the average intensity of light and the label of an image. The treatment also uses this one-dimensional summary, $\phi$, together with an additional (hidden) synthetic confounder, $U$. This is described by the following SCM:

$$\begin{cases} U \sim \text{Bern}(0.5), \\ X \sim \text{MNIST-image}(\cdot), \\ \phi := \left(\text{clip}\left(\frac{\mu_{N_x} - \mu_c}{\sigma_c}; -1.4, 1.4\right) - \text{Min}_c\right) \frac{\text{Max}_c - \text{Min}_c}{1.4 - (-1.4)}, \\ \alpha(\phi; \Gamma^*) := \frac{1}{\Gamma^* \, \text{sigmoid}(0.75\phi + 0.5)} + 1 - \frac{1}{\Gamma^*}, \\ \beta(\phi; \Gamma^*) := \frac{\Gamma^*}{\text{sigmoid}(0.75\phi + 0.5)} + 1 - \Gamma^*, \\ A \sim \text{Bern}\left(\frac{u}{\alpha(\phi; \Gamma^*)} + \frac{1-u}{\beta(\phi; \Gamma^*)}\right), \\ Y \sim N\big((2A - 1)\phi + (2A - 1) - 2\sin(2(2A - 1)\phi) - 2(2u - 1)(1 + 0.5\phi); 1\big), \end{cases} \tag{54}$$

where $c$ is a label of the digit from the sampled image $X$; $\mu_{N_x}$ is the average intensity of the sampled image; $\mu_c$ and $\sigma_c$ are the mean and standard deviation of the average intensities of the images with the label $c$; and $\text{Min}_c = -2 + \frac{4}{10}c, \text{Max}_c = -2 + \frac{4}{10}(c + 1)$. The parameter $\Gamma^*$ defines what factor influences the treatment assignment to a larger extent, i.e., the additional confounder or the one-dimensional summary. We set $\Gamma^* = \exp(1)$. For further details, we refer to (Jesson et al., 2021).

---

[9]After one-hot-encoding of categorical covariates.

For the experiments with HC-MNIST, we use a larger network size for our INFs (compared to other benchmarking experiments) to allow for more flexibility. We set the number of hidden units in fully-connected subnetworks to $h = 30$, and the dimensionality of representation $r = 30$. We also increase the number of training iterations to $n_{\text{iter},t} = 15,000$ and $n_{\text{iter},s} = 5000$.

Fig. 5 shows both ground-truth interventional ($\mathbb{P}(Y[a])$) and observational ($\mathbb{P}(Y \mid A = a)$) distributions together with our INFs A-IPTW estimator ($\hat{\mathbb{P}}^{\text{A-IPTW}}(Y[a])$). Remarkably, the interventional distributions in HC-MNIST are multi-modal and differ a lot from observational distributions.

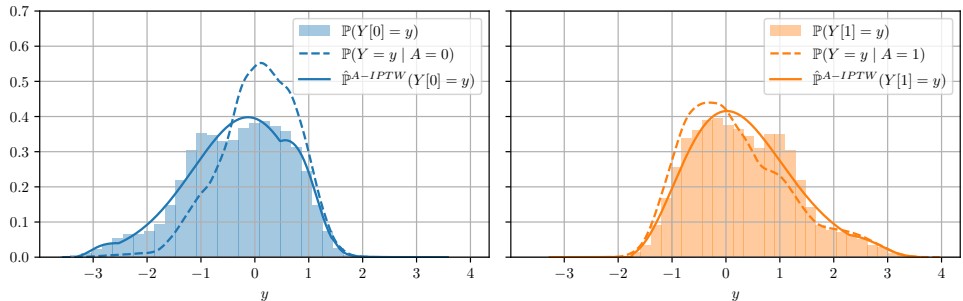

Figure 5: Empirical ground-truth interventional and conditional distributions of the HC-MNIST synthetic outcome. We also plot our INFs density estimator, i.e., $\hat{\mathbb{P}}^{\text{A-IPTW}}(Y[a])$.

# I   SYNTHETIC TWO-DIMENSIONAL DATA

In the following, we benchmark our INFs for estimating an interventional density of the multidimensional outcome, $d_Y = 2$.

**Noisy moons synthetic data.** We used a standard two-dimensional toy data generator, namely moons data.[10] It draws samples from two interleaving half-circles with different noise levels $\varepsilon$. The noise level controls the level of the overlap between two half-circles (a higher $\sigma$ corresponds to a better overlap, and, thus, a satisfaction of the positivity assumption). We drawing $n = 1000$ observations of two-dimensional covariates, i.e., $d_X = 2$, and use an inclusion to the top or bottom semi-circle as a treatment. Finally, we generate the synthetic outcome by rotating the covariates by a random treatment-specific angle, i.e., $\alpha_0$ and $\alpha_1$:

$$\begin{cases} X, A \sim \text{Make-Moons}(\cdot, \sigma), \\ \varepsilon \sim N(0; \sqrt{0.1}^2), \\ \alpha_0 = \frac{\pi}{4} + \varepsilon, \quad \alpha_0 = -\frac{\pi}{4} + \varepsilon, \\ Y := R(\alpha_1 A + \alpha_0 (1 - A))X + \varepsilon \mathbf{1}_2, \end{cases} \tag{55}$$

where $\mathbf{1}_2 = (1, 1)^T$, and $R(\alpha) = \begin{pmatrix} \cos \alpha & -\sin \alpha \\ \sin \alpha & \cos \alpha \end{pmatrix}$ is an $\alpha$-angle rotation matrix. We set $\sigma = 0.75$.

For the benchmarking with the noisy moons data, we increased the number of the training iterations for all the plug-in methods ($n_{\text{iter}} = 10000$) and for our INFs, ($n_{\text{iter},t} = 10000, n_{\text{iter},s} = 5000$). To model two-dimensional (conditional) density, we employed an auto-regressive extension of neural spline flows (Dolatabadi et al., 2020). We decreased the number of sampled points for approximating the cross-entropy, $K = 70$, to speed up the training, and set the number of knots for the student flow to $n_{\text{knots},s} = 5$.

**Results.** Table 6 shows the results. Here, our INFs (main) scores second best in terms of in-sample performance, but, more importantly, best in out-sample performance. MDNs, although scoring the best with in-samnple average log-probability, do not generalize well. Finally, we again confirmed, that our INFs are superior over their ablations and other existing methods, e.g., KDE and DKME.

Table 6: Results for synthetic experiments using the noisy moons synthetic data. The performance is benchmarked using the empirical in-sample / out-sample average log-probability for the two potential outcomes (i.e., $a = 0$ and $a = 1$). Reported: mean $\pm$ standard deviation over ten-fold train-test splits.

| | $a = 0$ | | $a = 1$ | |
|---|---|---|---|---|
| | log-prob$_{\text{in}}$ | log-prob$_{\text{out}}$ | log-prob$_{\text{in}}$ | log-prob$_{\text{out}}$ |
| TARNet* | $-2.907 \pm 0.121$ | $-3.005 \pm 0.263$ | $-2.781 \pm 0.092$ | $-2.955 \pm 0.222$ |
| MDNs | $\mathbf{-2.698 \pm 0.050}$ | $-2.887 \pm 0.173$ | $\mathbf{-2.683 \pm 0.051}$ | $-2.827 \pm 0.165$ |
| CNF [$\hat{=}$ INFs w/o stud flow] | $-2.767 \pm 0.087$ | $-2.935 \pm 0.239$ | $-2.807 \pm 0.162$ | $-2.900 \pm 0.183$ |
| KDE (Kim et al., 2018) | $-2.913 \pm 0.015$ | $-2.916 \pm 0.052$ | $-2.898 \pm 0.013$ | $-2.901 \pm 0.049$ |
| DKME (Muandet et al., 2021) | $-2.872 \pm 0.016$ | $-2.875 \pm 0.056$ | $-2.847 \pm 0.012$ | $-2.849 \pm 0.067$ |
| INFs w/o bias corr | $-2.787 \pm 0.057$ | $\underline{-2.794 \pm 0.130}$ | $-2.785 \pm 0.048$ | $\underline{-2.788 \pm 0.135}$ |
| INFs (main) | $\underline{-2.764 \pm 0.030}$ | $\mathbf{-2.766 \pm 0.102}$ | $\underline{-2.780 \pm 0.022}$ | $\mathbf{-2.785 \pm 0.134}$ |

Higher = better (best in bold, second best underlined)

---

[10]`https://scikit-learn.org/stable/modules/generated/sklearn.datasets.make_moons.html`

## J    Additional results

### J.1    IHDP dataset

Here, we provide additional results for the IHDP dataset with an alternative evaluation metric, that is, the empirical Wasserstein distance.

**Evaluation metric.** For one-dimensional outcomes, the Wasserstein distance between two distributions can be simply expressed via quantile functions

$$W^p(\mathbb{P}_1, \mathbb{P}_2) = \left( \int_0^1 |\mathbb{F}_1^{-1}(q) - \mathbb{F}_2^{-1}(q)|^p \, \mathrm{d}q \right)^{1/p}, \tag{56}$$

where $\mathbb{F}_1^{-1}(q)$ and $\mathbb{F}_2^{-1}(q)$ are quantile functions of $\mathbb{P}_1$ and $\mathbb{P}_2$, respectively. The Wasserstein distance is not upper-bounded and equals zero if and only if both distributions are the same. Here, we compute the empirical Wasserstein distance, i. e., $\hat{W}^1$, based on empirical quantile functions. This requires two samples: one from the ground-truth interventional distribution and another from the estimated density. Therefore, methods which do not provide direct sampling (e. g., KDE and DKME) cannot be used for evaluation.

Table 7 shows the results. Note that TARNet$^*$, i. e., the plug-in with the ground-truth conditional density estimator for this specific dataset due to the fact how the data was constructed. Hence, we do not interpret TARNet$^*$ as a baseline but rather interpret it as a bound for the best performance. We see that all baselines (MDNs, CNF, INFs w/o bias correction) are inferior by a large margin. In contrast, our INFs achieve a performance similar to the bound. In particular, our INFs perform overall best: our INFs are superior over the two other naïve plug-in estimators and the variant of INFs without bias correction. In sum, the results corroborate our findings from the main paper and add to the effectiveness of our INFs.

Table 7: Additional results for semi-synthetic experiments using the IHDP dataset. The performance is benchmarked using the empirical in-sample / out-sample Wasserstein distance (i.e., $\hat{W}^1_{\mathrm{in}}$ and $\hat{W}^1_{\mathrm{out}}$) for the two potential outcomes (i.e., $a = 0$ and $a = 1$). Reported: mean $\pm$ standard deviation over ten-fold train-test splits.

| | $a = 0$ | | $a = 1$ | |
|---|---|---|---|---|
| | $\hat{W}^1_{\mathrm{in}}$ | $\hat{W}^1_{\mathrm{out}}$ | $\hat{W}^1_{\mathrm{in}}$ | $\hat{W}^1_{\mathrm{out}}$ |
| TARNet$^*$ [$\hat{=}$ ground-truth for IHDP] | $0.048 \pm 0.014$ | $\mathbf{0.131 \pm 0.040}$ | $\mathbf{0.046 \pm 0.024}$ | $0.126 \pm 0.065$ |
| MDNs | $0.067 \pm 0.053$ | $0.156 \pm 0.054$ | $0.121 \pm 0.076$ | $0.183 \pm 0.071$ |
| CNF [$\hat{=}$ INFs w/o stud flow] | $0.118 \pm 0.048$ | $0.192 \pm 0.069$ | $0.111 \pm 0.087$ | $0.146 \pm 0.082$ |
| INFs w/o bias corr | $0.075 \pm 0.030$ | $0.137 \pm 0.051$ | $0.107 \pm 0.060$ | $0.128 \pm 0.057$ |
| INFs (main) | $\mathbf{0.040 \pm 0.009}$ | $0.132 \pm 0.051$ | $0.100 \pm 0.037$ | $\mathbf{0.117 \pm 0.055}$ |

Higher = better (best in bold, second best underlined)

### J.2    ACIC 2016 & 2018 datasets

In the following, we present detailed results of the experiments with ACIC 2016 and ACIC 2018 datasets. Fig. 6 reports the median performance for the individual datasets in ACIC 2016, and Fig. 7 for ACIC 2018. In the latter, datasets are grouped by sample size. We also show the performance gain of our INFs (when INFs score better than the baselines). The percentage of the datasets with the positive performance gain for our INFs roughly correspond to the percentage reported in the Table 3. For ACIC 2016, our INFs are the best method for 33 + 21 = 54 out of 2 * 77 = 154 potential outcomes of individual datasets (35%) with respect to out-of-sample average log-probability. For comparison, the second-best baseline (MDNs) are only best for 23 + 25 = 48 out of 2 * 77 potential outcomes of individual datasets (31%), and thus inferior. For ACIC 2018, our INFs are the best method for 8 + 9 out of 2 * 24 potential outcomes of individual datasets (35%). For comparison, the second-best baseline (also MDNs) are only best for 5 + 4 out of 2 * 24 potential outcomes of individual datasets (19%), and thus again inferior. This thus provides consistent performance that our INFs are highly effective.

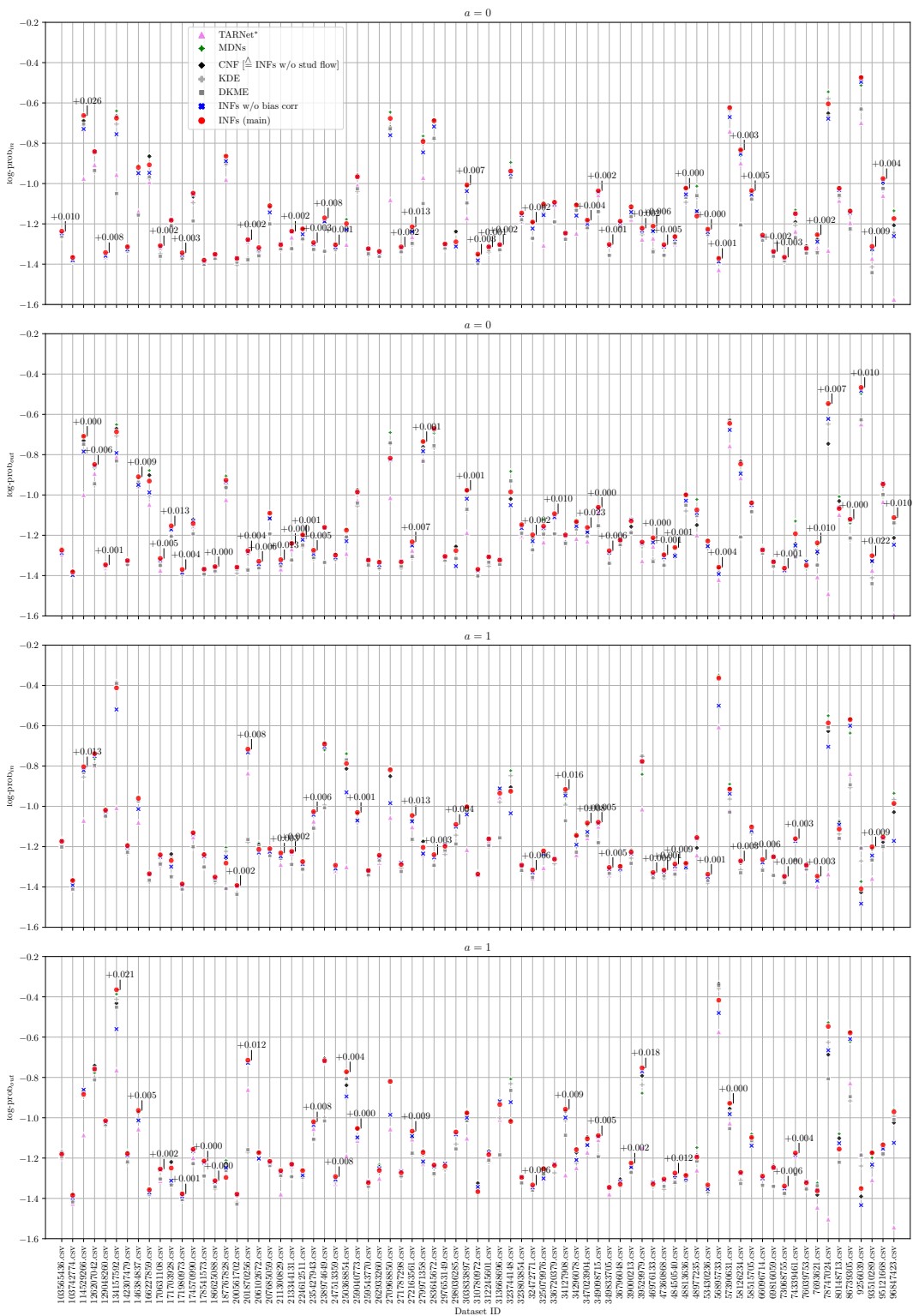

Figure 6: Detailed results for ACIC 2016. For each dataset, we perform five random train-test splits, tune the baselines on the first split, and evaluate the average in-sample / out-sample log-probability for each of the two potential outcomes separately. Shown: median over five runs and improvement of our INFs (main), when they score better than other baselines.

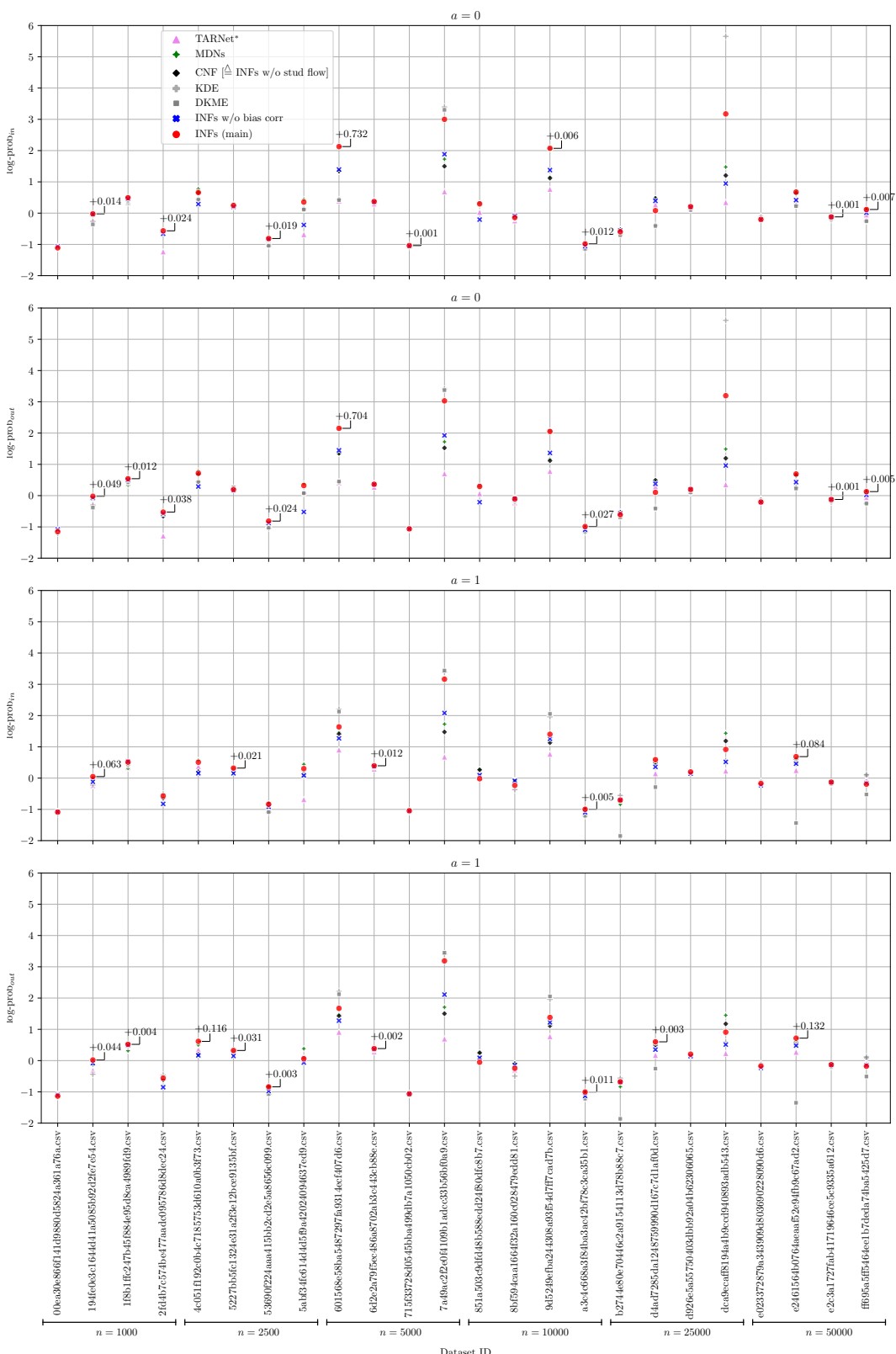

Figure 7: Detailed results for ACIC 2018, sorted with respect to sample sizes. For each dataset, we perform five random train-test splits, tune the baselines on the first split, and evaluate the average in-sample / out-sample log-probability for each of the two potential outcomes separately. Shown: median over five runs and improvement of our INFs (main), when they score better than other baselines.

# K    RUNTIME COMPARISON

INFs are a fully-parametric model and, therefore, provide a decent speed up at the inference time. This is particularly important for scalability, that is, for datasets with large sample size and high-dimensional covariates. In Table 8, we report the total runtime of the baselines and the different variants of our INFs. We see that the runtime of both full INFs and INFs (CA) stay relatively constant, but for the other baselines, it grows polynomially. This demonstrates the benefits of our INFs for scalability.

Table 8: Total runtime (in minutes) of the experiments using ACIC 2018 datasets with different sample sizes. Reported: mean $\pm$ standard deviation over four datasets and five runs for each size (lower is better). Experiments are carried out on Intel(R) Xeon(R) Silver 4316 CPU @ 2.30GHz.

| Sample size | 1000 | 2500 | 5000 | 10000 | 25000 | 50000 |
|---|---|---|---|---|---|---|
| TARNet[*] | $0.30 \pm 0.01$ | $0.33 \pm 0.01$ | $0.43 \pm 0.01$ | $0.64 \pm 0.03$ | $2.24 \pm 0.12$ | $8.16 \pm 0.22$ |
| MDNs | $0.29 \pm 0.05$ | $0.34 \pm 0.03$ | $0.54 \pm 0.05$ | $1.02 \pm 0.05$ | $5.06 \pm 0.71$ | $16.24 \pm 1.83$ |
| CNF [$\hat{=}$ INFs w/o stud flow] | $0.58 \pm 0.07$ | $0.82 \pm 0.11$ | $1.14 \pm 0.17$ | $2.33 \pm 0.23$ | $8.95 \pm 1.71$ | $29.38 \pm 5.55$ |
| KDE (Kim et al., 2018) | $0.52 \pm 0.09$ | $0.54 \pm 0.12$ | $0.55 \pm 0.07$ | $0.88 \pm 0.22$ | $1.70 \pm 0.22$ | $8.06 \pm 1.48$ |
| DKME (Muandet et al., 2021) | $0.03 \pm 0.01$ | $0.07 \pm 0.04$ | $0.13 \pm 0.05$ | $0.29 \pm 0.09$ | $2.28 \pm 0.35$ | $11.51 \pm 0.80$ |
| INFs w/o bias corr | $1.52 \pm 0.09$ | $1.48 \pm 0.08$ | $1.52 \pm 0.07$ | $1.55 \pm 0.08$ | $1.73 \pm 0.11$ | $1.73 \pm 0.08$ |
| INFs (main) | $2.47 \pm 0.07$ | $2.47 \pm 0.09$ | $2.48 \pm 0.08$ | $2.52 \pm 0.07$ | $2.59 \pm 0.09$ | $2.70 \pm 0.12$ |

## L   CASE STUDY: CALIFORNIA'S TOBACCO CONTROL PROGRAM

**Overview.** To show a real-world application of our INFs, we provide additional results using a case study where we evaluate the effect of California's tobacco control program (Abadie et al., 2010). This refers to the effect of Proposition 99, a large-scale tobacco control program introduced in California after 1988. Proposition 99 increased California's cigarette tax by 25 cents per pack, and earmarked the tax revenues to health and anti-smoking education. The main conclusion of Abadie et al. (2010) is that the effects of the tobacco control program are much larger than previously reported. The dataset has also found widespread use in causal inference ever since (e.g., Bellot & van der Schaar, 2021). In the original paper (Abadie et al., 2010), the results were based on a synthetic control method but without providing density estimates.

**Dataset.** After an initial pre-processing, the dataset consists of the 39 states, including California. For each state, we observe several covariates (e. g., beer consumption per capita, GDP per capita, retail price, and percent of people aged 15–24) and the outcome, i. e., cigarette sales per capita. These are recorded annually for each year from 1970 to 2001. Further details on the datasets are in (Abadie et al., 2010).

To apply our INFs, we make several gross assumptions. First, as there is only one treated state, it is impossible to satisfy the positivity assumption. Therefore, we consider a tuple (state, year) as an independent unit of measurement, thus obtaining $n = 1209$ observations with 12 treated observations (i. e., those of the state of California after 1989). We also add year as a covariate, which gives $d_X = 4 + 1$. We acknowledge that, even after the previous pre-processing, we still cannot formally guarantee the independence between units of measurement, as the observations of one state over time are not independent. Second, we assume the consistency holds, and there is no spillover effect between neighboring states, so that the potential outcome of one state is independent of the others.

**Results.** We plot the empirical conditional and the estimated interventional distributions in Fig. 8. The results go in line with the conclusion in (Abadie et al., 2010). Our main finding is that the introduction of the Proposition 99 ($a = 1$) to all the states from 1970 would substantially reduce tobacco sales. In particular, the mass of the interventional density is shifted to the left which accounts for the reduction of the consumption.

As a robustness check, we analyze the role of the smoothness hyperparameter. Our conclusion remains consistent if one specifies different smoothness hyperparameter for the student flow, i. e. $n_{\text{knots},s} = 5$ and 10. The specification of this hyperparameter is based on the prior knowledge of a researcher and cannot be chosen via observational data. However, we find consistent evidence of a positive effect.

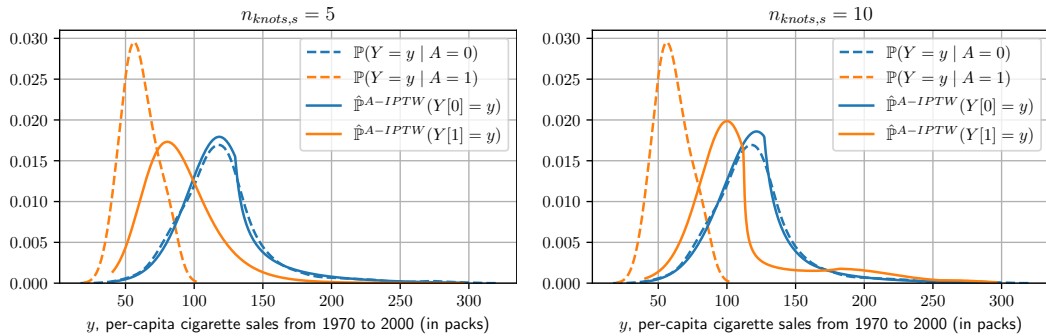

**Figure 8:** Empirical ground-truth conditional and estimated interventional distributions of cigarette sales per capita from 1970 to 2001. Treatment $a = 1$ corresponds to the introduction of the Proposition 99, that is, a comprehensive tobacco tax along with educational programs. We plot our INFs density estimator, $\hat{\mathbb{P}}^{\text{A-IPTW}}(Y[a])$ with different smoothness hyperparameter values $n_{\text{knots},s}$ of the student flow.

