# OpenReview forum: "Normalizing Flows for Interventional Density Estimation"
_ICLR.cc/2023/Conference — Submitted to ICLR 2023_

### Official Review · Reviewer_quXx · 2022-10-25

**Confidence:** 4
**Correctness:** 3
**Technical Novelty And Significance:** 4
**Empirical Novelty And Significance:** 3
**Recommendation:** 6

**Clarity, Quality, Novelty And Reproducibility:**

The paper is clearly written, has good quality and is novel. I am not sure about the reproducibility

**Strength And Weaknesses:**

### Strengths
1) The problem considered in the paper is interesting and the method is novel.
2) The paper is well-written with a good discussion of related works especially those related to interventional density estimation and normalizing flows for causal inference (Appendix A). I also appreciate the authors' attempt to make things clear for the readers by using colors in their formulas.
3) I think the experiment is quite extensive with various baselines for density estimation. The authors also compare different variants of their method.

### Weaknesses
1) The main weakness is the complexity of this method as in my opinion, it can be much simpler. For example, the student flow seems redundant as it just approximates $P(Y|do(A=a))$ though I know the authors want to avoid taking an average over $X$ during testing. In the ablation studies, the authors also consider a variant of the proposed method without the student flow, and when looking at the results in Table 2, I saw that this variant is not too bad compared to the full model.
2) Please correct me if I am wrong but I think we can use simple methods like Balancing Linear Regression [1], TARNet [2], DragonNet [3], … for interventional density estimation (IDE) problem in the paper. These methods are mainly designed for ITE estimation but we can easily modify their outputs to model $P(Y|X, A)$. Once we can model P(Y|X, A), we can simply estimate $P(Y|do(A)) = \mathbb{E}_{p(X)}[P(Y|X, a)]$ via backdoor adjustment. I would like to hear the authors’ opinions about this.
3) The authors should compare their method with existing ITE estimation methods to support their argument in the paper saying that “estimating the potential outcome density is better than just estimating the average potential outcome“.
4) In the TeacherFlow, I would like to understand why did the authors use $X, A$ to compute the weight $\theta$ of the conditional normalizing flow? Why don’t just model $P(Y|X, A)$ directly via a neural network with $X, A$ are inputs to this network? Since the distribution of outcomes $Y$ is usually simple. Do we really need a normalizing flow with hyper-network for its parameters?
5) The authors should provide more insights and explanations about why each component is useful for the method to work. Currently, they only use one metric, which is the log-likelihood, for comparison, which I think is not enough.

[1] Learning Representations for Counterfactual Inference, Johansson et al., ICML-2016

[2] Estimating individual treatment effect: generalization bounds and algorithms, Shalit et al., ICML-2017

[3] Adapting Neural Networks for the Estimation of Treatment Effects, Shi et al., NIPS-2019


**Summary Of The Paper:**

The paper considers an interesting problem in interventional causal inference, which is estimating the distribution of a potential outcome corresponding to an intervention. It proposes a fully parametric deep learning model for this task called Interventional Normalizing Flow that uses 2 flows.

The teacher flow computes the propensity score $\pi\_{a}(X)$ from $X$ and parameters of a conditional normalizing flow network (CNF) from $(X, A)$. This conditional flow network maps Gaussian noise to $P(Y|X, A)$.

The student flow is a set of unconditional normalizing flows for each intervention $A=a$. Each unconditional flow maps Gaussian noise to $g(Y, \beta\_a)$ where $\beta\_a$ is the set of parameters corresponds to $A=a$. Each unconditional flow w.r.t. $A=a$ is learned by mimicking $P(Y|do(A=a))$ obtained from the teacher flow. This problem is transformed into an equivalent problem of solving moment conditions. This paper also introduces an (optional) one-step bias correction for doubly robust estimation of $\beta\_a$ by leveraging the propensity score prediction from the teacher flow.


**Summary Of The Review:**

Overall, this is a good paper which proposes a novel method to address an interesting problem. The only problem is that the model is more complex than it should be. This can limit the practical applications of the work. I give this a paper a weak accept.

---

> ### Author Response · Authors · 2022-11-15
> **Response to Reviewer quXx**
>
> Thank you for your positive review and your helpful comments! We addressed all of them in our revised paper.
> ### Response to “Weaknesses”
> 1. Thanks for raising the important question of whether there is a simpler way to solve our problem. In particular, the question is whether the student flow network is redundant, but it is not. In fact, the student flow is crucial for computational performance. Although the student flow has a similar estimation performance in terms of goodness-of-fit, it has constant inference time (e.g., during the evaluation phase) and does not depend on the dimensionality of covariates and the size of the training data. This is a major advantage of parametric treatment effect estimators over semi-paramentric plug-in estimators. For a detailed runtime comparison, we refer to Appendix K. In summary, the student flow allows our method to scale well on large datasets, which are common nowadays in many application areas such as medicine. After reading your question, we came to the realization that we need to discuss the importance of the student flow in greater depth. Hence, we added an additional explanation throughout our paper (see “Introduction” and “Experiments” Sections).
> 2. Thank you for asking this question, which gives us the opportunity to highlight the differences between existing methods and ours. The methods you mentioned are designed for standard regression tasks (i.e., estimating the conditional mean) and not for conditional density estimation. Hence, these methods are _not_ valid baselines for our INFs. Using them for density estimations requires non-trivial adaptions, such as replacing standard neural networks with normalizing flows. In fact, the architecture of our teacher network is inspired by DragonNet. Note that we could also employ a TARNet-like architecture instead, which would perform better on data where the propensity score does not share much structure with the conditional outcome densities (see Curth and van der Schaar  2021). However, we refrained from implementing such a model: The aim of our paper is not to compare TARNet and DragonNet-type architectures (well studied in the literature), but rather to adapt neural networks for efficient estimation of interventional densities. \
> Nevertheless, we made additional efforts to develop a new variant of TARNet for the purpose of density estimation. For this, the natural way is – in our opinion – to model the conditional outcome distribution as normal with a constant standard deviation and a conditional mean, given by the output of the TARNet. This would allow us to extend TARNet (which we call TARNet*) by simply adding one nuisance parameter of standard deviation (could be found via maximum likelihood together with other parameters). We report it as another baseline but find that it is inferior (see the updated Section “Experiments” and Appendix F.1). Note that such a method has a key limitation in that the conditional outcome distribution is assumed to be normal (or another distribution of choice), which is likely to be violated in practice. Further, the conditional normal distribution is not a universal density estimator, but our INFs are.
> 3. We would like to emphasize that estimating interventional densities is an inherently **different** task than estimating ITEs/ ATEs. In many applications, methods which capture uncertainty or model density are of higher importance than only yielding a point estimate of the mean  (Spiegelhalter, 2017; van der Bles et al., 2019). Here, ITE/ ATE methods **cannot** be applied as they aim at estimating the mean and do not assume any density model. Hence, they do **not** yield density estimates without non-trivial adaptations (see also point 2). We thus refrain from using ITE methods as baselines in our experiments.
> 4. In our paper, we use _conditional_ normalizing flows (i.e., universal density estimators) because we do not want to make any assumption about the distribution of $Y$. Standard NFs can only estimate _unconditional_ distributions (Rezende & Mohamed, 2015). To estimate conditional distribution, it is standard practice to parametrize the NF by a hypernetwork  (Trippe & Turner, 2018). In fact, hypernetwork architectures are state-of-the-art for neural conditional density estimation and are also used for MDNs and VAEs. We added an explanation of the above to Appendix F.1 “Naїve semi-parametric plug-in estimators” and to the Section “INFs”.
> 5. We followed your suggestion and added a discussion about each component to the revised paper (see “Experiments” Section). Regarding the performance evaluation: The log-likelihood is the only metric available to compare all methods because some methods do not allow for direct sampling. For methods which support sampling (including ours), we also provided results using empirical Wasserstein distance in Appendix J.1. In our revised paper, we also added detailed results for the individual ACIC datasets as a new Appendix J.2.

---

### Official Review · Reviewer_uUEZ · 2022-10-25

**Confidence:** 3
**Correctness:** 4
**Technical Novelty And Significance:** 2
**Empirical Novelty And Significance:** 4
**Recommendation:** 8

**Clarity, Quality, Novelty And Reproducibility:**

The paper is mostly clear, some background on influence functions and doubly robust estimation can make the paper stronger. The theoretical framework of the presented method existed before, the main novelty lies in the algorithmic instantiation and the performance of the presented methods. The information in the paper sounds sufficient for the reproduction of the results, however, I did not try to reproduce the results myself.

**Strength And Weaknesses:**

**Strengths**
* The presented methods fill a gap in the literature on estimating interventional densities using parametric models. Therefore the problem considered is timely and significant.
* The motivation is clear as the example in Fig. 1 is representative of why the problem considered is important.
* I found the schematic in Fig. 2 very helpful, it provides a nice summary of the components of the model.
* The presentation and the logical flow of the paper are clear, I specifically appreciated the use of colors in the text for determining what corresponds to nuisance parameters and what does not.
* The background contains a nice summary of the existing literature for estimating interventional effects for the case of backdoor adjustment. The literature has focused on average treatment effect (ATE) and the non-ATE literature is mostly non-parametric or non-algorithmic.

**Comments**
* The problem of estimating $\mathbb{E}_{X \sim P(X)} \big[ P(Y|X, A) \big]$ given samples from $X, Y, A$ sounds like a well studied estimation problem in statistics regardless of the context of interventional distributions (here, the interventional or counterfactual data is not used and we are interested in the above estimation problem). To me, it is surprising that Normalizing Flows are not used for this purpose before so I cannot confirm the novelty of the methodology. Given that (conditional) normalizing flows, the theory of semi-parametric estimation of the interventional distribution, doubly robust estimation tools existed before the main contribution seems to lie in the addition of the student network, and framing and solving the optimization problem.

* Given that the work is applicable only to the backdoor adjustment, even for the cases in which we can identify the effect of intervention we cannot use this method. However, the extension to the more general case sounds feasible. This would largely broaden the impact and scope of the paper. Can the authors discuss the possibility of this extension?

* What happens if $Y$ is not one-dimensional? It seems that the method relies on creating a grid on $\mathcal{Y}$. Is this applicable and does this scale to higher dimensional $Y$ variables?

* Can the authors explain why in the case of the synthetic dataset in Fig. 3 KDE is doing very well (even better than the proposed method) for $a=0$ but fails for $a=1$? Where does this asymmetry come from?

* Can the authors include the SCMs used for other examples in the supplementary?

* I think the contributions of this paper by themselves are enough for the acceptance of the paper if these techniques are not used in the context of causal inference and interventional density estimation before. I am happy to change my score to acceptance if the authors include the above information.

* Can the authors include the famous tobacco control example in the experiments?

* To make this a standalone paper, I encourage the authors to include a short summary of the influence functions and doubly robust estimation. More specifically, to me, it wasn't quite clear how we go from Eq. 5 to 7, 8.


**Post rebuttal**

I thank the authors for clarifying their contributions and distinguishing between the original estimation theory and the semi-parametric giving rise to the optimization framework and the suggested student-teacher architecture. Now it's clear why the classical density estimation techniques aren't as powerful as semi-parametric methods and why finite-dimensional estimands are not applicable. I think the new additions (adding more background on semi-parametric theory and SCMs, including the SCMs associated with the experiments, adding results on the tobacco control experiment) make the paper more standalone and powerful. Hence I changed my score to acceptance.

**Summary Of The Paper:**

The paper introduces an optimization framework with a model consisting of a teacher and student normalizing flows for estimating the interventional distribution for the case of backdoor adjustment. The developed optimization uses one-step bias correction and utilizes influence functions for doubly robust estimation of the interventional density. Results on multiple synthetic and semi-synthetic datasets are reported showing that INFs outperform nonparametric and ablated methods.

**Summary Of The Review:**

The paper has good quality, including further information (summarized under comments) can make it stronger.

---

> ### Author Response · Authors · 2022-11-15
> **Response to Reviewer uUEZ**
>
> Thank you for your detailed and constructive review! We took all your comments at heart and improved our paper accordingly.
>
> ### Response to “Comments”
>
> * We agree with you that the estimation problem $\mathbb{E}_{X \sim P(X)} \big[ P(Y|X, A) \big]$ can be solved in a well-established manner by plugging in an estimator $\hat{P}(Y|X, A)$ (=plug-in approach, this is included in our baselines). However, efficiency theory results for this estimation problem are relatively new (Kennedy et al., 2021). This is due to the fact that the interventional density is a _functional target estimand_, but standard nonparametric efficiency theory only yields results for _finite-dimensional estimands_ (such as the average treatment effect). Kennedy et al. (2021) were the first to introduce the finite-dimensional projection parameter (Eq. 3) and derive the moment condition for its efficient estimation. Our main contribution is indeed centered around framing the efficiency result from Kennedy et al. (2021) as an optimization problem and tailoring the student network to solve it. The extension of ours is non-trivial: the optimization objective (A-IPTW) consists of several terms (cross-entropy and one-step bias correction), and each of them has to be effectively evaluated and then used for back-propagation. This is different from Kennedy et al. (2021), who propose to solve a system of nonlinear equations numerically to obtain an efficient estimator. As such, ours is the first fully-parametric deep learning method to estimate the density of interventions. Note also that we are the first to adapt normalizing flows for this purpose, even though they have been used for (classical) density estimation. We clarified in our Introduction that this is our main contribution.
> * Standard treatment effect estimation literature relies on three main assumptions, which are enough for the use of backdoor adjustment (van der Laan 2006, Shalit 2017, Wager 2018). More complex adjustment rules (e.g., front-door adjustment for napkin graphs) have the following limitations: (1) they require more unusual, complex assumptions which are often violated in practice; and (2) they require a complex efficient estimation theory (Vowels 2022). This could be an interesting direction for future research but it is beyond the scope of our paper. We added this to our discussion (see new paragraph “Comparison to other identification strategies”, Appendix B.2 “Causal model and indentification”, in our revised paper).
> * We used a grid for one-dimensional outcomes because this resulted in a better convergence of our model. For multidimensional outcomes, we can replace our cross-entropy loss with maximum likelihood based on the sampled data from the teacher flow. We extended our equations introducing the cross-entropy and conditional cross-entropy losses to show that our INFs are applicable to multidimensional outcomes. Also, we introduced a new 2D benchmark based on a toy moons dataset (see Appendix I, "Synthetic two-dimensional data"), where our INFs again demonstrate their effectiveness.
> * We used the median heuristic to choose the smoothing parameter for KDE. This is standard for kernel-based methods (Garreau 2017). It seems like the median heuristic provided a valid smoothing parameter for a=0, but did oversmoothing for a=1.
> * Thanks for this suggestion. We added the SCMs for IHDP and HC-MNIST to Appendix H “Dataset details”. For ACIC, they are vastly different for every dataset. Here, we refer to the original documentation [1,2].
> * We followed your suggestion and included the California tobacco control example as a new case study (see new Appendix L “Case Study” and the new subsection in the Results). Importantly, our method goes in line with the result in (Abadie et al., 2010). Our INFs suggest that the introduction of California’s tobacco tax program ($a=1$) to all the states would substantially reduce tobacco sales.
> * We followed your suggestion and added a summary regarding nonparametric efficiency theory and influence functions (see new Appendix B.3 “Efficiency theory and influence functions”).
>
> [1] [https://jenniferhill7.wixsite.com/acic-2016/competition](https://jenniferhill7.wixsite.com/acic-2016/competition)
>
> [2] [https://www.synapse.org/#!Synapse:syn11294478/wiki/486304](https://www.synapse.org/#!Synapse:syn11294478/wiki/486304)

---

### Official Review · Reviewer_A7uk · 2022-10-27

**Confidence:** 3
**Correctness:** 4
**Technical Novelty And Significance:** 3
**Empirical Novelty And Significance:** 3
**Recommendation:** 5

**Clarity, Quality, Novelty And Reproducibility:**

The authors did extensive experiments and showed their methods could outperform other methods. But the authors can improve the introduction to audiences unfamiliar with causal inference and the normalizing flow method.

**Strength And Weaknesses:**

To improve the readability, the authors may want to:
1. Add more introduction to the structural causal model.
2. In the example provided in Introduction Section,  explain interventional distributions, observational distributions, and counterfactual distributions. Explicitly define the notation Y [a].


Question: Why does Table 3 compare % of runs with best performances, while other tables compare the log-probability? And why are the columns for Table 3 named log-prob?


**Summary Of The Paper:**

The authors propose a fully-parametric, deep-learning method for interventional density estimation, called Interventional Normalizing Flows (INFs). INFs provide a properly normalized density estimator. The authors further develop a two-step training procedure with a one-step bias correction for efficient and doubly robust estimation.

**Summary Of The Review:**

The method proposed in this work is novel. The authors estimate the density of potential outcomes after interventions from observational data and propose a fully-parametric deep learning method INF. The authors also provide extensive numerical studies to support the performance of INF.

---

> ### Author Response · Authors · 2022-11-15
> **Response to Reviewer A7uk**
>
> Thank you for your review! We appreciate that you found our method novel and significant.
>
> ### Response to “Strength and weaknesses”
> We addressed your comments regarding the presentation of our paper and are confident that this improved both clarity and readability.
>
> 1. We added additional information regarding our structural causal model to Appendix B.2 “Causal model and indentification”. In particular, we included references to standard causal inference literature such as (Pearl 2009; Bareinboim et al., 2022).
> 2. We added an explanation of the differences between observational, interventional, and counterfactual distributions to our Introduction. We have further added the definition of the potential outcomes $Y(a)$ to both the Introduction and our new Appendix B.2 “Causal model and indentification”.
>
> ### Response to “Question”
> Table 3 reports the results for multiple datasets that contain the same covariates but different synthetic outcomes. All the datasets have a different maximum possible log-probability and therefore cannot be averaged together. Instead, we report the percentage of datasets where the corresponding methods perform best. To avoid misunderstandings and improve clarity, we changed the column names from “in-/out log-prob” to “in- / out- % best”. Furthermore, we added detailed results for the individual datasets as a new Appendix J.2 “ACIC 2016 & 2018 datasets”.
>
>
> ### Response to “Clarity, Quality, Novelty and Reproducibility”
> We now provide additional background materials. Specifically, we now offer an introduction to causal inference (see new Appendix B.2 “Causal model and indentification”). Furthermore, we now provide an introduction to normalizing flows (see new Appendix B.1 “Normalizing flows”). We are confident that our paper is now readable by a much broader audience.

---

### Author Response · Authors · 2022-11-15
**Response to all reviewers**

Thank you very much for your positive evaluation of our paper and your helpful feedback! We addressed all of your comments in our revised paper (see rebuttal PDF) and highlighted changes in 🔵blue color🔵. Key changes are:
- We improved the presentation of our paper. In particular, we added a background on structural causal models, normalizing flows, and nonparametric efficiency theory.
- We performed additional experiments using more ACIC 2016 datasets, added a case study of the famous California tobacco experiment (Abadie et al., 2010), and included a new baseline (i.e., an extended version of TARNet for density estimation). Our method performs consistently superior, thereby confirming its effectiveness.
- We highlighted the novelty of our method: ours is the first deep learning method for estimating the density of interventional outcomes using machine learning. To achieve this, we are the first to leverage normalizing flows and efficiency theory from  Kennedy et al. (2021).

Given the improvements above, we are confident that our paper is now a valuable contribution to the research field and a good fit for ICLR 2023.

---

### Author Response · Authors · 2023-01-31
**Post-review response**

Thank you all for the positive and constructive feedback! We are glad to see that “the presented methods fill a gap in the literature on estimating interventional densities using parametric models” and that the reviews found our method “novel”,  “interesting, and “timely and significant”.

We thank for the constructive feedback that we received on the initial version of our manuscript. As a result, we improved our manuscript in several ways:
1. **Clarified novelty and contributions**. We have now clarified the novelty of our method and highlighted our contributions. Importantly, our work is the first to propose a fully-parametric, deep-learning method for estimating interventional densities. We further explained why deriving a fully-parametric method like ours is non-trivial and of large significance.
2. **Stronger motivation**. We have greatly expanded our motivation why estimating the density of potential outcomes is important for medical decision-making. Thereby, we highlight that our work fills an important gap in the literature.
3. **New scalability study**. We added new results to demonstrate the scalability of our method. Specifically, we included a two-dimensional setting. For high-dimensional settings, existing baselines become completely impractical due to memory and time constraints. In contrast, our proposed method can still solve them, which is a major advantage.
4. **Additional baseline**. We included an additional baseline method, i.e., an extended TARNet, which models the mean of a conditional homoscedastic normal distribution. We find that our proposed method is consistently superior.
5. **New results**. We added new results from ~100 experiments from the ACIC datasets. Across all experiments, we find that our proposed method is clearly superior.
6. **New case study**. We included a new case study to show the value of our method for decision-making in medical practice. For this, we followed suggestions from the reviewers and added new experiments using data from famous California’s tobacco control program and discussed how our results expand over existing estimates.
7. **More background materials**. We have greatly expanded our background materials. Therein, we now provide an extensive introduction to normalizing flows, causal modelling, causal identification, efficiency theory, and influence functions.
We are confident that our revised paper is now of substantially better quality (the new and significantly revised version can be found on arXiv).

We would like to stress that our paper makes **important and novel contributions**: (1) We formulated a novel tractable A-IPTW estimation objective. Importantly, our objective is different from the moment condition proposed by Kennedy. (2) We introduced a fully-parametric, deep learning method for interventional density estimation based on normalizing flows. This requires a careful architecture based on two entangled normalizing flows that are learned via novel two-step training procedure. (3) We demonstrated that our method scales well to both large and high-dimensional datasets. Our method outperforms existing non- and semi-parametric methods by a large margin. Our method further has constant inference time (during the evaluation phase regardless of the data size). This is a major advantage of our fully-parametric method over existing baselines.

We would like to kindly disagree with the AC that our method employs ‘standard practice’ in causal inference. On the contrary, we make **non-trivial contributions**: (1) The interventional density itself is an infinitely-dimensional target estimand, unlike standard ATE or ATT. Hence, standard efficiency theory is not applicable. While there is a theory for semi-parametric estimation of interventional densities, a flexible instantiation in the form of a deep learning method is missing. This presents one of our novelties. (2) The majority of efficient treatment effect estimation methods require a single computation step given nuisance parameters (e.g., by averaging). This is different from our task where we have to solve a complex optimization problem. (3) We derive a tractable optimization objective and then propose a novel two-step training procedure (see Algorithm 1 in our paper). This is different from other two-step training procedures, e.g., for ITE, as the optimization objective includes averaging and integration and cannot be represented as a regression on pseudo-outcomes.  Nevertheless, we understand that our initial manuscript was not sufficiently clear in how we presented our novelty. To address this, we have now made a thorough revision. As a result, we now state our contributions more clearly. Here, the new analyses from above were of great help.

Also, as a response to the AC, we conducted additional experiments with the trapezoidal quadrature rule and found that the results remain similar and without noticeable improvement. We added the results and a discussion to our paper.

---

### Decision · Program_Chairs · 2023-01-20

**Decision:**

Reject

**Justification For Why Not Higher Score:**

The theory comes straight out existing papers, and the adaption of normalizing flows to this setup is not that surprising. I appreciate the response given to reviewers on the novelty aspect, but I have to say I had a hard time appreciating it. For instance, the names "teacher" and "student" networks basically amount to using a black-box for nuisance parameters and another black-box for the causal parameters, a very standard practice. Here, it happens to be a normalizing flow, which requires some extra thought steps concerning numerical computations, but even then they are not too sophisticated (e.g. the solution in (9) is just a direct discretization, not even a quadrature method is attempted)...

**Justification For Why Not Lower Score:**

N/A

**Metareview: Summary, Strengths And Weaknesses:**

The paper adapts normalizing flows to the task of estimating densities of potential outcomes.

Strengths: very logical exposition, sensible methods, experiments include many other baselines.

Weaknesses: the theory comes straight out existing papers, and the adaption of normalizing flows to this setup is not that surprising. I appreciate the response given to reviewers on the novelty aspect, but I have to say I had a hard time appreciating it. For instance, the names "teacher" and "student" networks basically amount to using a black-box for nuisance parameters and another black-box for the causal parameters, a very standard practice. Here, it happens to be a normalizing flow, which requires some extra thought steps concerning numerical computations, but even then they are not too sophisticated (e.g. the solution in (9) is just a direct discretization, not even a quadrature method is attempted)...